# ST3GAL1 is a target of the SOX2-GLI1 transcriptional complex and promotes melanoma metastasis through AXL

Silvia Pietrobono [1], Giulia Anichini[1,2], Cesare Sala [3], Fabrizio Manetti[4], Luciana L. Almada[5], Sara Pepe[1,2], Ryan M. Carr[5], Brooke D. Paradise[5], Jann N. Sarkaria [6], Jaime I. Davila [7], Lorenzo Tofani[8], Ilaria Battisti[9], Giorgio Arrigoni [9,10], Li Ying[11], Cheng Zhang[12], Hu Li [12], Alexander Meves [11], Martin E. Fernandez-Zapico[5] & Barbara Stecca [1]✉

Understanding the molecular events controlling melanoma progression is of paramount importance for the development of alternative treatment options for this devastating disease. Here we report a mechanism regulated by the oncogenic SOX2-GLI1 transcriptional complex driving melanoma invasion through the induction of the sialyltransferase ST3GAL1. Using in vitro and in vivo studies, we demonstrate that ST3GAL1 drives melanoma metastasis. Silencing of this enzyme suppresses melanoma invasion and significantly reduces the ability of aggressive melanoma cells to enter the blood stream, colonize distal organs, seed and survive in the metastatic environment. Analysis of glycosylated proteins reveals that the receptor tyrosine kinase AXL is a major effector of ST3GAL1 pro-invasive function. ST3GAL1 induces AXL dimerization and activation that, in turn, promotes melanoma invasion. Our data support a key role of the ST3GAL1-AXL axis as driver of melanoma metastasis, and highlight the therapeutic potential of targeting this axis to treat metastatic melanoma.

[1] Core Research Laboratory – Institute for Cancer Research and Prevention (ISPRO), Viale Pieraccini 6, 50139 Florence, Italy. [2] Department of Medical Biotechnologies, University of Siena, Viale M. Bracci 16, 53100 Siena, Italy. [3] Department of Clinical and Experimental Medicine, University of Florence, Viale Morgagni 50, 50134 Florence, Italy. [4] Department of Biotechnology, Chemistry and Pharmacy, University of Siena, Via A. Moro 2, 53100 Siena, Italy. [5] Schulze Center for Novel Therapeutics, Division of Oncology Research, Department of Oncology, Mayo Clinic, Rochester, MN 55905, USA. [6] Department of Radiation Oncology, Mayo Clinic, Rochester, MN 55905, USA. [7] Department of Health Sciences Research, Mayo Clinic, Rochester, Rochester, MN 55905, USA. [8] Department of Neurosciences, Psychology, Drug Research and Child Health, University of Florence, Viale Pieraccini 6, 50139 Florence, Italy. [9] Proteomics Center, University of Padova and Azienda Ospedaliera di Padova, Via G. Oris 2B, 35129 Padova, Italy. [10] Department of Biomedical Sciences, University of Padova, Via U. Bassi 58B, 35131 Padova, Italy. [11] Department of Dermatology, Mayo Clinic, Rochester, MN 55905, USA. [12] Department of Molecular Pharmacology & Experimental Therapeutics, Mayo Clinic, Rochester, MN 55905, USA. ✉email: b.stecca@ispro.toscana.it

Malignant melanoma is the most aggressive and treatment-resistant form of skin cancer. Its aggressiveness is based on the highly metastatic potential of melanoma cells even at early stage of the disease. Although recent treatment advances using targeted therapy and immune checkpoint inhibitors achieved long-term survival in a small subset of patients with advanced melanomas, metastatic melanoma still remains an incurable disease[1]. Therefore, identification and targeting of key drivers of melanoma metastasis are crucial steps towards an effective control of tumor progression.

Aberrant glycosylation, in particular increased sialylation, is an important determinant of malignant phenotype, as it directly impacts on key processes supporting tumor progression and metastasis, including cell adhesion, motility, invasion, and immune evasion[2–7]. Enhanced sialic acid levels increase resistance to apoptosis and modulate the function of immune cells[8], as well as alter tumor cell–cell interaction, promoting cell detachment from a site of origin. Because sialylated glycoconjugates regulate adhesion and promote motility, they may also be important for the colonization and metastatic potential. Although aberrant glycosylation has been associated with melanoma progression[9,10], the biological significance and mechanism(s) controlling this post-translational modification during disease pathogenesis remain for the most poorly understood.

Recent studies from our and other groups have shown an interplay between the transcription factor GLI1, the final effector of the Hedgehog (HH) pathway, and the pluripotency transcription factor SOX2 in several types of cancer including melanoma[11–13]. Here we identify a set of genes co-regulated by SOX2-GLI1 contributing to aberrant O-glycosylation in metastatic melanoma. Using in vitro and in vivo assays, we demonstrate that ST3GAL1, a β-galactoside-α-2,3-sialyltransferase-1 that catalyzes the transfer of sialic acid from cytidine monosphosphate (CMP)-sialic acid to galactose-containing substrates, is a crucial driver of melanoma invasion and metastasis downstream the SOX2-GLI1 transcriptional complex. In addition, biochemical and functional studies reveal that the receptor tyrosine kinase AXL is a major mediator of the pro-invasive effects of ST3GAL1 in melanoma. Our study reveals a functional ST3GAL1-AXL axis driving melanoma metastasis and suggests that inhibition of this axis may become in the future a promising therapeutic strategy for metastatic melanoma.

## Results

### O-glycosylation is a common regulated pathway of SOX2 and GLI1.
To identify genes co-regulated by both SOX2 and GLI1, we performed whole-transcriptome analysis in patient-derived SSM2c cells, which were obtained from a subcutaneous metastasis[11,14], after genetic silencing of either SOX2 or GLI1 with specific shRNAs (Fig. 1a). Three pairs of total RNAs from each condition were generated for library preparations and subsequent RNA-sequencing (RNA-seq). All samples had over 50 million reads with over 93% of the reads mapping to the human genome (obtained from the UCSC Genome Browser assembly ID: hg19). Using differential expression analysis (fold change (FC) > 1.5, false discovery rate (FDR) < 0.1 as standard cut-offs), we identified 1104 entities commonly altered in both SOX2- and GLI1-knocked down cells (Fig. 1b and Supplementary Data 1). Grouping of genes into enrichment clusters revealed by DAVID functional annotation tool (https://david.ncifcrf.gov/) identified O-Glycan biosynthesis pathway ($p = 5.9E-3$) among the top-7 pathways significantly downregulated in both SOX2-and GLI1-depleted melanoma cells (Fig. 1c). As aberrant glycosylation has been associated with melanoma progression and immunosuppression[4,9,10,15], we examined in detail the expression

of genes that are components of the O-linked glycan biosynthetic pathway. RNA-seq results coupled with quantitative real-time PCR (qPCR) validation identified six different glycosyltransferases significantly downregulated in both SOX2- and GLI1-silenced cells: the N-acetylgalactosaminyltransferases GALNT3, GALNT6 and GALNT12, the sialyltransferase ST3GAL1 and the N-acetylglucosaminyltransferases GCNT1 and GCNT2 (Fig. 1d–f). Data mining of transcriptomic datasets from independent clinical cohorts revealed consistently higher levels of ST3GAL1 in melanoma compared to nevi and in metastatic cases compared to primary melanomas (Supplementary Fig. 1). Western blotting analysis confirmed a strong reduction of ST3GAL1 protein level in both SOX2- and GLI1-depleted A375 M6 and SSM2c cells (Fig. 1g), and a consistent increase of ST3GAL1 after ectopic expression of these transcription factors (TFs; Fig. 1h). Based on these results, we focused on defining the functional role of the sialyltransferase ST3GAL1 in melanoma metastasis.

### ST3GAL1 expression correlates with melanoma progression.
The β-galactoside-α-2,3-sialyltransferase-1 ST3GAL1 catalyzes the transfer of sialic acid from cytidine monosphosphate (CMP)-sialic acid to galactose-containing substrates and is associated with cancer progression and drug resistance[16,17]. Data mining of publicly available transcriptomic datasets of several types of skin cancer (GSE7553)[18] revealed that ST3GAL1 is preferentially upregulated in malignant melanomas compared to basal cell carcinoma (BCC) or squamous cell carcinoma (SCC; Fig. 2a). In addition, a transcriptomic dataset of human nevi, primary, and metastatic melanomas (GSE46517)[19] showed that ST3GAL1 is strongly associated with melanoma progression, with higher ST3GAL1 mRNA levels in primary melanomas compared to nevi and in metastatic compared to primary melanomas (Fig. 2b). Moreover, in silico data analysis showed that ST3GAL1 is altered in 23% of human melanomas (missense mutations, gene amplification, or mRNA upregulation; Fig. 2c). Consistently with transcriptomic data, immunohistochemical analysis of human melanoma tissue microarrays showed higher proportion of metastatic cases with medium/high ST3GAL1 staining compared to primary melanomas ($p = 0.0198$) or nevi ($p = 0.0007$; Fig. 2d, e and Supplementary Table 1). ST3GAL1 staining in melanoma cells was observed in the cytoplasm with a perinuclear granular pattern (Fig. 2d, high magnification). In human normal skin ST3GAL1 staining was confined to sparsed cells with weak cytoplasmic expression (Fig. 2d). Accordingly, western blot analysis showed that ST3GAL1 was expressed at various levels in metastatic melanoma cell lines, but barely present in normal human epidermal melanocytes (NHEM; Fig. 2f). Altogether, these data indicate that ST3GAL1 is consistently expressed in human melanomas and is associated with tumor progression. These results prompted us to investigate the role of ST3GAL1 in the pathobiology of melanoma and address its functional regulation by SOX2 and GLI1.

### ST3GAL1 regulates melanoma invasiveness.
To investigate the role of ST3GAL1 in melanoma progression, we silenced ST3GAL1 in a lung metastatic clone derived from A375 cells (A375 M6)[20] and in SSM2c cells using two independent ST3GAL1 shRNAs (LV-shST3GAL1.1 and LV-shST3GAL1.2), which target the 3′-UTR or the CDS region of ST3GAL1 respectively (Fig. 3a). We first assessed whether ST3GAL1 plays a role in melanoma cell invasiveness using in vitro migration and invasion assays. In the scratch assay, motility and wound closure of scratched area were significantly reduced in ST3GAL1 depleted cells in a time-dependent manner, with a rate of reduction of 30–40% in LV-shST3GAL1.1 and LV-shST3GAL1.2 A375 M6

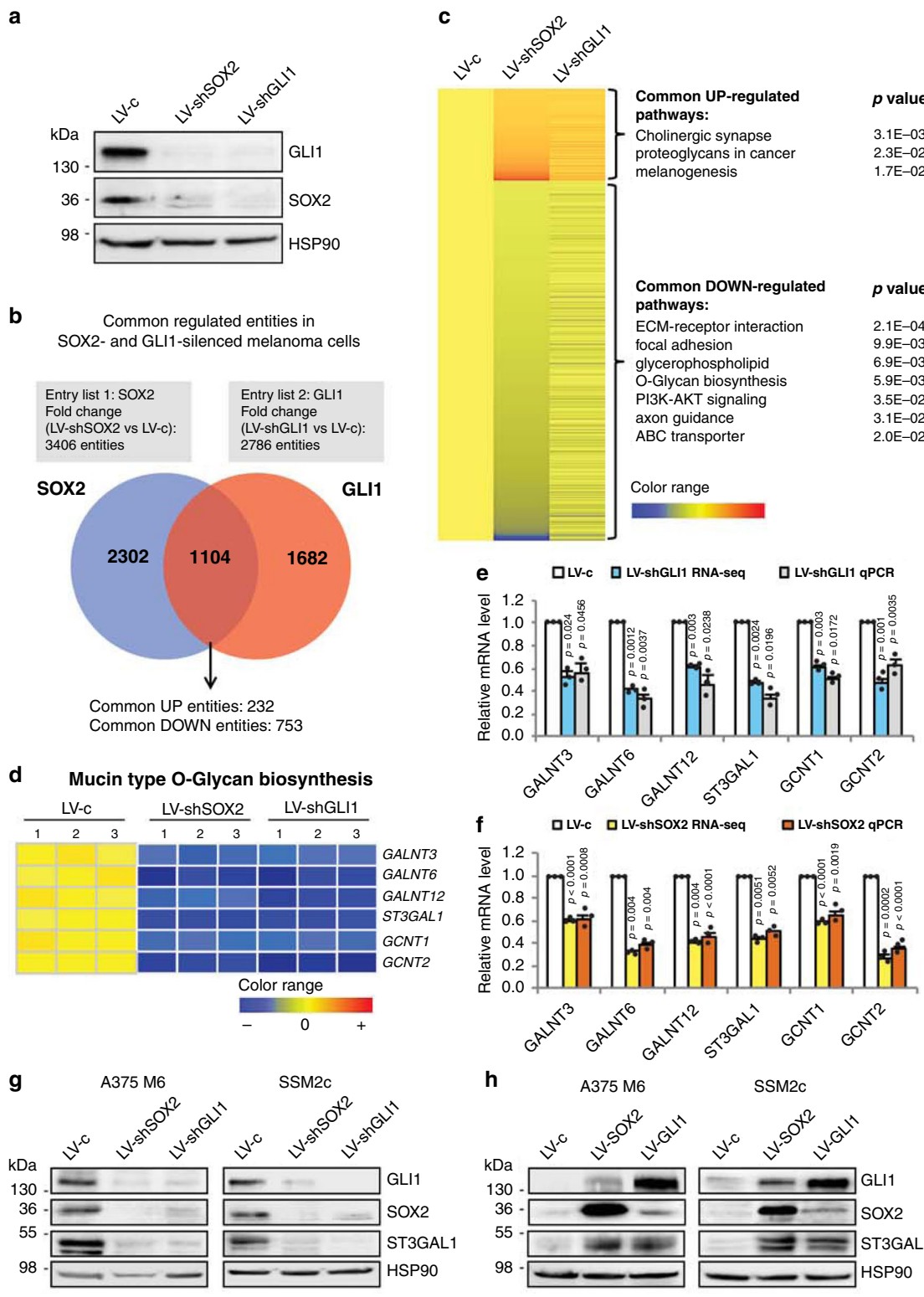

cells compared to scramble, respectively (Fig. 3b), and of ~80–90% in LV-shST3GAL1.1 and LV-shST3GAL1.2 SSM2c cells (Fig. 3c). In addition, ST3GAL1 knockdown strongly reduced the ability of A375 M6, or nearly abrogated that of SSM2c cells, to invade through Matrigel-coated Boyden chambers in a transwell invasion assay (Fig. 3d, e). Expression of the EMT markers N-cadherin, Vimentin, SNAI1 (Snail), and SNAI2 (Slug) decreased upon ST3GAL1 depletion, suggesting that ST3GAL1 regulates the transition to a more mesenchymal phenotype (Fig. 3a).

Consistent with these results, ectopic expression of ST3GAL1 in the metastatic cell line MeWo and in patient-derived metastatic cells M51 (Fig. 3f), both of which express low levels of ST3GAL1 (Fig. 2f), accelerated wound closure and enhanced invasion in vitro (Fig. 3g–j), with concomitant increased expression of EMT markers (Fig. 3f). Modulation of ST3GAL1 expression had no significant effect on melanoma cell growth, although ST3GAL1 overexpression slightly decreased the fraction of early apoptotic cells only in MeWo cells, without affecting late

**Fig. 1 Transcriptomic analysis of genes cooperatively regulated by SOX2 and GLI1 in melanoma cells. a** Western blot analysis of SOX2 and GLI1 in patient-derived melanoma cells (SSM2c) used for RNA-seq. Cells were transduced with LV-c (scrambled control), LV-shSOX2 (shRNA targeting SOX2), or LV-shGLI1 (shRNA targeting GLI1). **b** Venn diagram showing intersection of common up- or downregulated entities in patient-derived melanoma cells knockdown for either SOX2 or GLI1 (fold change >1.5, FDR < 0.1 as standard cut-offs for differential expression analysis). **c** Heat Map of hierarchically clustered genes in SSM2c cells transduced with LV-c, LV-shSOX2, or LV-shGLI1. The expression level of all entities is shown as a mean of triplicate samples. Significantly upregulated (red) and downregulated (blue) Gene Ontology terms and enrichment *P* values by DAVID (functional annotation clustering) are shown. **d** Genes that are components of the mucin type O-glycan biosynthesis pathway in LV-c, LV-shSOX2, or LV-shGLI1 melanoma cells are shown in detail. **e, f** Validation of RNA-seq results with qPCR of genes shown in **d**. Data are expressed as fold change relative to scrambled cells (LV-c), which were equated to 1. Gene expression was normalized relative to *TBP* and *HPRT* housekeeping genes and expressed as mean ± s.e.m. *P* value was calculated by two-tailed unpaired Student's *t* test (n = 3 biological independent samples). **g, h** Western blot analysis of SOX2, GLI1, and ST3GAL1 in A375 M6 and SSM2c cells transduced with LV-c, LV-shSOX2, or LV-shGLI1 (**g**) or with LV-c, LV-SOX2, or LV-GLI1 (**h**). Blots in **a**, **g**, and **h** are representative of n = 3 biological independent experiments. HSP90 was used as loading control. Source data are provided as Source Data file.

apoptosis (Supplementary Fig. 2). Altogether these results indicate that ST3GAL1 endows melanoma cells with metastatic potential.

**ST3GAL1 is required for in vivo melanoma metastasis.** We first examined how depletion of ST3GAL1 impacts on the ability of melanoma cells to survive and proliferate in a metastatic environment, using a model based on intracardiac instillation of melanoma cells[10]. We injected A375 M6 cells stably transduced with lentiviral particles carrying one of the two independent shRNA against ST3GAL1 (LV-shST3GAL1.1 or LV-shST3GAL1.2) or scramble control (LV-c) and a luciferase reporter vector (Fig. 4a). Effective ST3GAL1 silencing with both shRNAs was confirmed by qPCR and western blotting (Fig. 4b, c). We performed intracardiac injection of transduced A375 M6 in athymic nude mice and assessed for colonization by in vivo bioluminescence imaging (BLI) at days 7 and 14 post-injection. A successful intracardiac injection was indicated on day 1 by images showing systemic bioluminescence distributed throughout the animal. Mice injected with either LV-shST3GAL1.1 or LV-shST3GAL1.2 displayed significantly reduced seeding ability compared to mice injected with LV-c in multiple organs (Fig. 4d–f). Mice injected with A375 M6 cells expressing both ST3GAL1 shRNAs showed fewer and much smaller metastatic lesions in the lungs (Fig. 4g, h).

We next determined whether modulation of ST3GAL1 affects metastasis formation in vivo using a xenograft model of metastasis. We injected A375 M6 cells stably transduced with GFP/luciferase reporter and LV-shST3GAL1.1, LV-ST3GAL1 or scramble control (LV-c) into the flank of athymic nude mice, and monitored mice for primary tumor growth, presence of circulating melanoma cells (CTCs) in the blood, and metastasis (Fig. 5a, b). We observed no significant difference in tumor growth between LV-c, LV-shST3GAL1.1, and LV-ST3GAL1, except a slight tumor growth delay in mice injected with A375 M6 cells transduced with LV-shST3GAL1 (Fig. 5c). When local tumors reached the same tumor volume (21–24 days post-injection; Fig. 5c, d), primary tumors were surgically resected, and the ability of cells to enter the blood stream (CTCs) and give rise to distant metastases was examined. Silencing of ST3GAL1 substantially reduced the frequency of circulating melanoma cells in the blood (Fig. 5e) and strongly inhibited the metastatic potential of A375 M6 cells, as shown by the reduced distribution of luminescence in organs such as lungs (Fig. 5f, g). Forty-five days post-injection mice were killed and lungs and main organs dissected for histology and immunohistochemistry. Mice injected with LV-shST3GAL1 exhibited significantly reduced number of lung micrometastases compared to LV-c (Fig. 5h, i). Ectopic expression of ST3GAL1 did not affect the number of CTCs nor the number of micrometastases per lung compared to control (Fig. 5e–i), although mice injected with

melanoma cells transduced with LV-ST3GAL1 exhibited higher number of macrometastases (Supplementary Fig. 3). These data provide evidence that ST3GAL1 silencing drastically impairs the ability of aggressive melanoma cells to enter the blood stream, colonize distal organs and seed, survive and proliferate in the metastatic environment.

**SOX2 and GLI1 co-regulate *ST3GAL1* gene transcription.** Both SOX2 and GLI1 positively modulate ST3GAL1 expression (Fig. 1), thus we sought to investigate whether the effect of SOX2 and GLI1 was a result of a direct transcriptional control on *ST3GAL1*. In silico analysis of ST3GAL1 regulatory regions (RRs) (obtained from the UCSC Genome Browser assembly ID:hg38) using the TFBIND bioinformatics software (http://tfbind.hgc.jp) identified several putative SOX2- and GLI-binding sites (BS) possessing the reported consensus sequences (SOX2-BS: wwTGnwTw)[21] (GLI-BS: GACCACCCA)[22,23]. Chromatin immunoprecipitation assay (ChIP) of SOX2 and GLI1 followed by qPCR using different sets of primers spanning the putative BS within the selected RRs revealed co-occupancy of SOX2 and GLI1 at a distal enhancer element located at ~10 kb upstream the transcription start site (TSS) of ST3GAL1 (enhancer II), with a 3–8-fold enrichment in ST3GAL1 signal over ChIP with non-specific IgG in two different melanoma cell types (Fig. 6a–c). Meanwhile, negative control and primers spanning *ST3GAL1* promoter showed no significant enrichment (Fig. 6b, c). To confirm the ability of SOX2 and GLI1 to control ST3GAL1 expression, we cloned the ST3GAL1 enhancer II construct encompassing putative SOX2-BS and GLI-BS (−12,453/−11,457 bp from TSS; Fig. 6d and Supplementary Fig. 4) into a luciferase reporter vector, and tested the effects of SOX2 and GLI1 on transcriptional activation of ST3GAL1 in SSM2c cells. We observed a strong induction of the luciferase activity upon ectopic expression of SOX2, which was reverted after depletion of GLI1, and viceversa (Fig. 6e, f). As GLI1 directly regulates the expression of SOX2[11] and SOX2 appears to modulate that of GLI1 (Fig. 1g, h), we assessed whether the observed induction of ST3GAL1 transcription could depend on a sequential regulation of the two TFs expression by overexpressing SOX2 in cells depleted for GLI1 and viceversa. Luciferase assays showed that SOX2 overexpression partially compensates for the absence of GLI1 (Supplementary Fig. 5a), similarly to what happens after ectopic expression of GLI1 in SOX2-silenced cells (Supplementary Fig. 5b), suggesting that both TFs are required for the induction of ST3GAL1 transcription. This was also confirmed at both mRNA and protein levels (Fig. 6g and Supplementary Fig. 5c–f). Next, disruption of the putative GLI-BS (−12,187/−12,178 bp from TSS) or of the two putative SOX2-BS (SBS1 at −12,212/−12,204 bp and SBS2 at −11,944/−11,936 bp from TSS) through site-directed mutagenesis (Fig. 6d) further corroborates our hypothesis of a direct transcriptional regulation of *ST3GAL1*

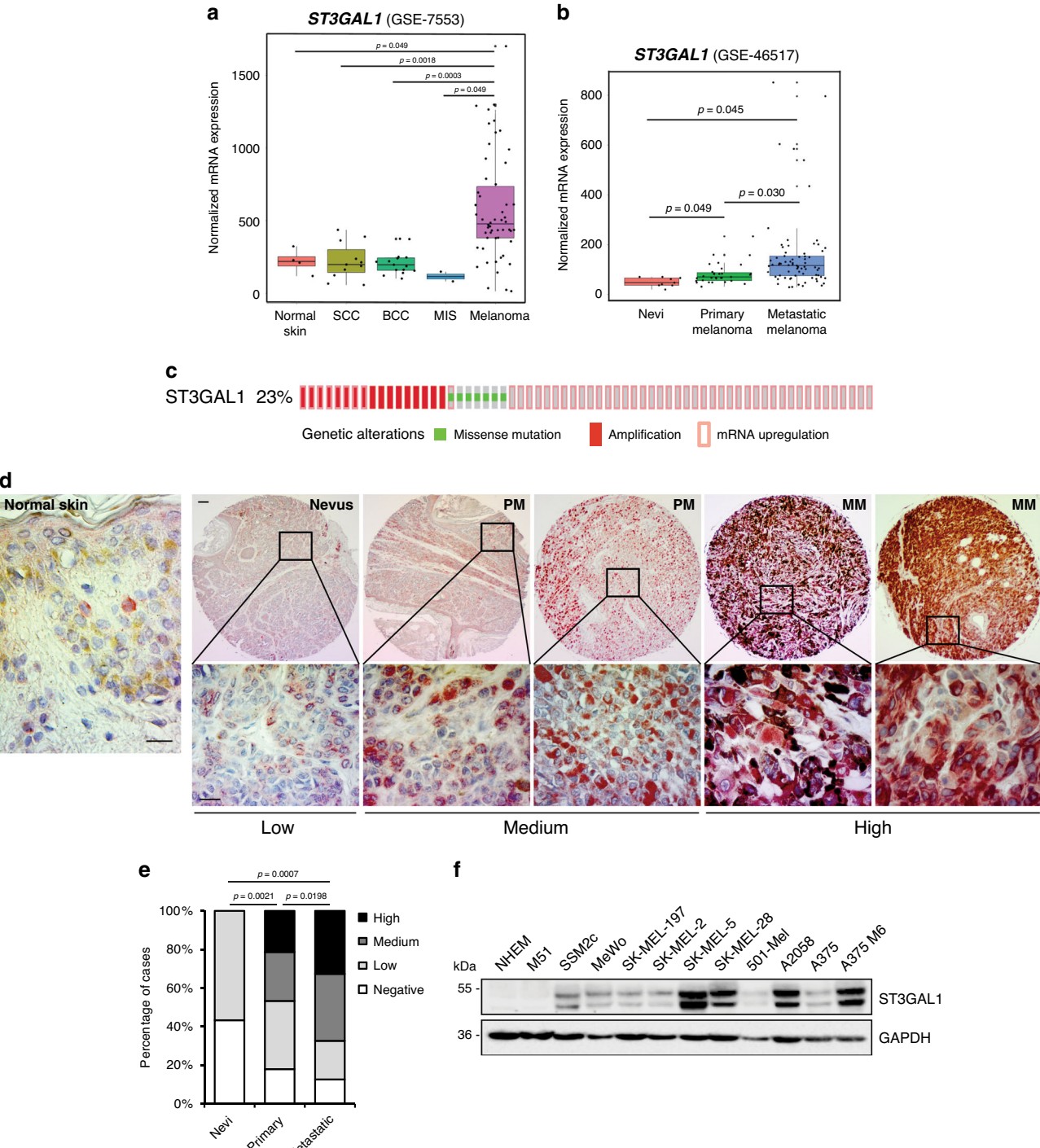

by both SOX2 (through its binding at SBS2) and GLI1 (Fig. 6h). The slight transactivation of ST3GAL1 GBSmut enhancer II by GLI1 was completely abrogated when GLI1 was overexpressed in cells depleted for SOX2 (Supplementary Fig. 5g), similarly to that of SBS2mut enhancer II by SOX2 (Supplementary Fig. 5h). However, while SOX2-induced transcriptional activation of ST3GAL1 occurs even in absence of a functional GBS, disruption of SBS2 appeared to reduce transactivation by GLI1 (Fig. 6h). This effect was still abrogated when GLI1 was overexpressed in SOX2-depleted cells (Supplementary Fig. 5g), suggesting that GLI1 regulates ST3GAL1 both directly and indirectly through SOX2. In support of this regulation, a positive correlation

between the expression of SOX2 and ST3GAL1 mRNA was observed in a panel of commercial and patient-derived melanoma cells (Supplementary Fig. 6).

Next, we asked whether ST3GAL1 could mediate SOX2- and GLI1-induced melanoma invasiveness. We first silenced ST3GAL1 in A375 M6 cells overexpressing either SOX2 or GLI1. Our results showed that both GLI1 and SOX2 increased melanoma cell invasion and that depletion of ST3GAL1 was able to suppress the effects of both SOX2 and GLI1 (Fig. 6i–k). Consistently, ectopic expression of ST3GAL1 in SOX2- or GLI1-depleted SSM2c cells was able to rescue the decrease in melanoma cell invasion produced by SOX2 or GLI1 silencing

**Fig. 2 Expression of ST3GAL1 in cutaneous melanomas and association with melanoma progression. a, b** Box plots illustrating expression of *ST3GAL1* mRNA in skin cancers (**a**) and in nevi, primary and metastatic melanoma samples (**b**). Box-plots report median (central lines), 25th and 75th percentiles (box limits), and upper and lower whiskers represent values no further than x1.5 interquartile range (IQR). Data were obtained from the analysis of the public available microarray data sets GSE7553 and GSE46517, respectively. In GSE7553: normal skin (*n* = 4), in situ melanomas (MIS) (*n* = 2), primary and metastatic melanomas (*n* = 54), basal cell carcinomas (BCC) (*n* = 15), and squamous cell carcinomas (SCC) (*n* = 11). In GSE46517: nevi (*n* = 9), primary melanomas (*n* = 31), metastatic melanomas (*n* = 73). *P* value was calculated by ANOVA and Holm-Sidak's test. **c** Genomic profile of *ST3GAL1* in melanoma patients obtained from Skin Cutaneous Melanoma data set (TCGA, Provisional) using cBioportal database (http://www.cbioportal.org)[60,61]. In all, 23% of melanoma samples present alterations of *ST3GAL1*, including gene amplification, mRNA upregulation or somatic mutations. **d** Immunostaining of ST3GAL1 in human melanoma tissue microarrays, including nevi (*n* = 23), primary (*n* = 56), and metastatic (*n* = 40) malignant melanomas. Representative images of human normal skin, nevi, primary (PM), and metastatic melanomas (MM). ST3GAL1 staining was evaluated blindly and scored as negative (no signal or present in <5% of cells), low, medium, or high (see Supplementary Table 1 for details and clinico-pathological characteristics of samples). Sections were counterstained with hematoxylin. Top panels show representative cores at low magnification, lower panels are high magnification of top images. Scale bars 100 μm. **e** Quantification of ST3GAL1 staining for each score (negative, low, medium, and high) in benign nevi, primary, and metastatic melanomas as shown in **d**. *P* values compare medium and high staining in each group and were calculated by ANOVA and Tukey's test (*n* = 23 nevi, *n* = 56 primary melanomas, *n* = 40 metastatic melanomas). **f** Western blotting of ST3GAL1 in normal human epidermal melanocytes (NHEM) and melanoma cell lines. GAPDH was used as loading control. Blots are representative of *n* = 3 biological independent experiments. Source data are provided as Source Data file.

(Supplementary Fig. 5i–k). Overall, these data reveal ST3GAL1 as a key downstream mediator that contributes to the aggressive behavior of melanoma cells induced by SOX2 and GLI1.

**AXL mediates the pro-invasive effects of ST3GAL1 in melanoma.** To identify putative mediators of ST3GAL1 in metastatic melanoma cells, we performed proteomic analysis of sialylated proteins. To this end, we prepared whole cell extracts of A375 M6 transduced with either LV-ST3GAL1 or LV-shST3GAL1.1 and enriched for sialylated proteins using MAL lectin affinity chromatography (Fig. 7a). Mass spectrometric analysis led to the identification of more than 300 proteins, 43 of which showed a significant increased sialylation level (fold change ≥1.5, *p* value < 0.05) in the enriched fraction of ST3GAL1 sample (Supplementary Data 2). After removing contaminant unglycosylated proteins (i.e. ribosomal proteins, tubulins, and heat shock proteins), gene ontology analysis with DAVID functional annotation tool revealed enrichment in biological processes relevant to metastasis such as cell motility, cell migration, and locomotion control mechanisms (Fig. 7b). Proteins identified as MAL-bound that are involved in the aforementioned cellular processes are shown in Supplementary Table 2. To validate our proteomic analysis, we examined the sialylated state of integrin β4 (ITGB4), that has been recently reported as ST3GAL1-sialylated target[24], and of 3 receptor tyrosine kinases (RTKs) that have been linked to melanoma progression: AXL receptor (also known as UFO)[25], nerve growth factor receptor (NGFR) (known also as CD271)[26], and epidermal growth factor receptor (EGFR)[27,28], that are expressed in melanoma cells at variable levels (Supplementary Fig. 7a). Immunoprecipitation with MAL-II lectin, which specifically recognizes α2,3-linked sialic acid residues, followed by western blot showed increased integrin-β4, AXL, NGFR, and EGFR levels in ST3GAL1 expressing cells (Fig. 7c; in agreement with our proteomic analysis). Remarkably, ST3GAL1 did not affect sialylation of integrin-α5, as already reported[24], corroborating the reliability of our MS results (Fig. 7c and Supplementary Data 2).

We next asked whether the increased sialylation of the identified RTKs by ST3GAL1 could be relevant for their biological activation in melanoma. We employed the human phospho-RTK assay in A375 M6 to profile the relative phosphorylation levels of 49 different activated transmembrane proteins that become phosphorylated at tyrosine residues in their catalytic domain, such as AXL, EGF, and NGF receptor family proteins. Ectopic expression of ST3GAL1 increased phosphorylation status of AXL compared to scrambled LV-c cells, without altering that of EGFR or NGFR (Fig. 7d). This was confirmed by immunoprecipitation of AXL, EGFR, or NGFR followed by

immunoblotting with anti-phospho-Tyrosine (Fig. 7e). Further, ectopic expression of ST3GAL1 in melanoma cells increased the phosphorylation of AXL at residue Tyr702 in the catalytic kinase domain (Fig. 7f), which becomes autophosphorylated in response to AXL activation[29], suggesting that ST3GAL1-mediated increased sialylation of AXL could be responsible for its activation.

To further understand whether sialylation could induce AXL dimerization and activation, we stimulated serum-starved A375 M6 cells with Gas6 and analyzed AXL dimerization. AXL appeared to exist mainly as monomer, whereas Gas6 stimulation induced its partial dimerization only in presence of endogenous ST3GAL1 (Fig. 7g). Consistent with the dimerization result, A375 M6 cells depleted for ST3GAL1 showed no AXL phosphorylation at Tyr702 in response to ligand stimulation (Fig. 7h). These data suggest that sialylation induces AXL dimerization and activation, likely increasing the affinity of AXL for its ligand Gas6.

The AXL receptor kinase is known to play a critical role in cell proliferation, survival, cancer stem cell maintenance, and metastasis in several cancers[25], and has been associated with melanoma cell motility, invasion, and drug resistance[30–32]. Thus, it represents a reasonable candidate for mediating the pro-invasive effects of ST3GAL1. As expected, ectopic expression of ST3GAL1 in A375 M6 and MeWo cells triggered a significant increase in melanoma cell invasion. Of note, concomitant silencing of AXL in these cells was able to fully revert the pro-invasive effects mediated by ST3GAL1 (Fig. 8a–d), as demonstrated by the reduction of Vimentin and TFs SNAI1 and SNAI2 (Fig. 8a), in contrast to what observed after depletion of either NGFR (Supplementary Fig. 7b–d) or EGFR (Supplementary Fig. 7e–g). In addition, overexpression of AXL or its stimulation by Gas6 enhanced the invasive phenotype only in presence of endogenous ST3GAL1 (Supplementary Fig. 8), suggesting that sialylation represents the key effector of this pathway. AXL impacted the migratory ability of A375 M6 ST3GAL1-overexpressing cells but not that of MeWo cells, suggesting that additional mediators could be involved in this process, possibly NGFR (Supplementary Fig. 9). Overall, our data suggest that sialylated AXL is a major mediator of the pro-metastatic role played by ST3GAL1 in melanoma.

To further support the relevance of the SOX2/GLI1-ST3GAL1-AXL axis in melanoma progression, we performed single-cell analysis in cells derived from melanoma brain metastasis PDX models (M12, M15, and M27) as well as normal melanocyte cultures (Supplementary Figs. 10 and 11). These data show that *ST3GAL1, GLI1, SOX2,* and *AXL* are co-expressed in melanoma

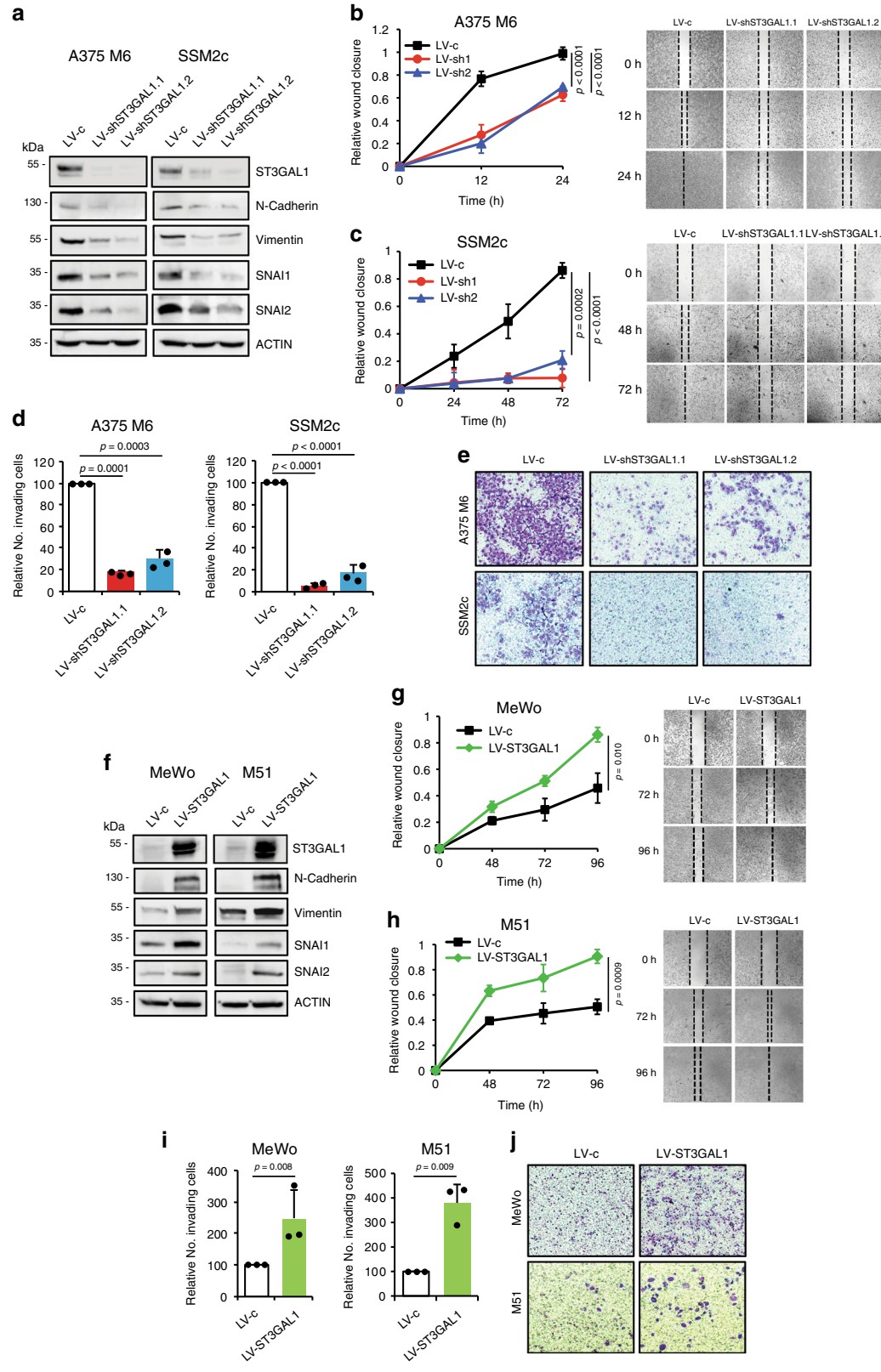

cells with similar or higher levels in tumor samples (M) vs. normal (N), except M12 (Fig. 8e–g and Supplementary Data 3). Furthermore, protein levels of GLI1, SOX2, ST3GAL1, and AXL were analyzed in short-term cultures of primary and metastatic melanomas. Western blots showed that primary melanoma cells express lower levels of SOX2, GLI1, ST3GAL1, and AXL than metastatic melanoma cells, and that the latters harbor a stronger activation of SOX2/GLI1-ST3GAL1-AXL axis (Fig. 8h). In addition, analysis of a cohort of 481 human melanoma samples from The Cancer Genome Atlas (TCGA) showed that expression

**Fig. 3 ST3GAL1 regulates melanoma cell migration and invasiveness. a** Western blot analysis of ST3GAL1 and EMT markers in A375 M6 and SSM2c melanoma cells stably expressing shRNA targeting ST3GAL1 (LV-shST3GAL1.1 or LV-shST3GAL1.2) or scrambled control (LV-c). **b, c** Scratch assay (wound closure) in A375 M6 and SSM2c cells transduced as indicated. Data are represented as mean ± s.d. *P* value was calculated by ANOVA and Tukey's test (*n* = 3 biological independent experiments). **d** Matrigel invasion assay in A375 M6 and SSM2c cells transduced as indicated. Data represent mean ± s. d and are expressed as fold change relative to control (LV-c), which was equated to 100. *P* value was calculated by ANOVA and Tukey's test (*n* = 3 biological independent experiments). **e** Representative images of **d. f** Western blot analysis of ST3GAL1 and EMT markers in MeWo and M51 melanoma cells transduced as indicated. **g, h** Scratch assay (wound closure) in MeWo (**g**) and M51 (**h**) cells transduced as indicated. **i** Matrigel invasion assay in MeWo and M51 cells transduced as indicated (*n* = 3 biological independent experiments). Data are expressed as fold change relative to control (LV-c), which was equated to 100. **j** Representative images of **i**. In **g, h,** and **i** data are represented as mean ± s.d and *P* value was calculated by two-tailed unpaired Student's *t* test. Blots in **a** and **f** are representative of *n* = 3 biological independent experiments. ACTIN was used as loading control. Source data are provided as Source Data file.

## Discussion

of *ST3GAL1* is significantly increased in BRAF mutant melanomas compared to wild type cases, and that of *AXL* is increased in TP53 mutant melanomas (Supplementary Fig. 12). These data support the existence of both non-genetic (the GLI1-SOX2 transcriptional mechanism) and genetic mechanisms controlling the ST3GAL1-AXL axis.

## Discussion

Malignant melanoma is one of the most lethal skin cancers worldwide and one of the malignancies with the highest potential to metastasize. Sialylation, the addition of sialic acid from a donor substrate to terminal positions of glycoprotein and glycolipid carbohydrate groups, has been shown to play major roles in tumor cell invasiveness and tumor progression. Here we show that the sialyltransferase ST3GAL1 is transcriptionally regulated by both GLI1 and SOX2 in melanoma and is a crucial driver of melanoma cell invasiveness and melanoma metastasis in vivo. Our work indicates that the receptor tyrosine kinase AXL is a major mediator of the pro-invasive effects of ST3GAL1 in melanoma (Fig. 8i). Our study suggests that targeting ST3GAL1-linked processes may become in the future a promising therapeutic approach to prevent or treat established melanoma metastasis.

Aberrant glycosylation is one of the hallmarks of cancer with altered gene expression signatures of sialyltransferases. Mounting evidence demonstrated that the alteration of sialylation and the levels of sialyltransferase activities are relevant for cancer invasion and metastasis[33]. ST3GAL1 has been shown to exert a tumor promoting effect in a PyMT mouse model of breast cancer[34]. ST3GAL1 is crucial for the maintenance of glioblastoma growth, through the regulation of stemness traits[16]. Furthermore, ST3GAL1 promotes cell migration and invasion, and confers resistance to paclitaxel in ovarian cancer[17]. However, no data are available on the role of ST3GAL1 in melanoma.

Our study indicates that ST3GAL1 expression levels correlate with melanoma progression. Indeed, we found that ST3GAL1 is expressed at low levels in normal skin and nevi, and its expression increases during melanoma progression at both mRNA and protein levels. Both microarray dataset and immunohistochemistry show that ST3GAL1 expression is significantly higher in metastatic compared to primary melanomas. In addition, ST3GAL1 is preferentially expressed in human melanomas compared to other types of skin cancer, such as BCC or SCC, which is consistent with the higher potential to metastasize of melanomas respect to other types of skin cancer.

Genetic silencing of ST3GAL1 reduces melanoma cell migration and invasion in vitro, whereas ST3GAL1 overexpression has the opposite effects. Importantly, we show that ST3GAL1 silencing in a highly invasive melanoma cell line reduces the number of melanoma cells able to enter the blood stream (circulating melanoma cells) and drastically diminishes metastatic dissemination to the lungs. In addition, ST3GAL1 depletion inhibits seeding of melanoma cells in several organs, suggesting that sialylation by ST3GAL1 may be required for the ability of melanoma cells to enter the blood stream, form metastases, and survive in a metastatic environment. In agreement with the identified role of ST3GAL1 in melanoma progression, ST3GAL1 is expressed at higher levels in metastatic compared to primary melanoma cases. Together our in vitro and in vivo data point toward several promoting effects of ST3GAL1 in melanoma progression, and provide strong evidence that silencing of ST3GAL1 inhibits the metastatic process in vivo. These effects appear not related to growth advantage, because modulation of ST3GAL1 expression does not impact on the growth of subcutaneous melanoma xenografts.

Our data also indicate that overexpression of ST3GAL1 does not endow melanoma cells with increased ability to metastatize. This might be explained with that fact that level of endogenous ST3GAL1 is sufficient to explicate its function. However, we found that melanoma cells overexpressing ST3GAL1 give rise to larger metastases compared to control cells, suggesting that melanoma cells with high ST3GAL1 expression may have a better chance to strive and survive in "host" tissues.

Here we identified the transcription factors SOX2 and GLI1 as positive regulators of ST3GAL1. This is in agreement with previous studies showing that SOX2 and GLI1 are both involved in melanoma progression. For instance, SOX2 has been shown to contribute to melanoma metastasis[35] and to be highly expressed in melanomas compared to nevi[36]. In addition, SOX2 regulates survival, self-renewal, and tumorigenicity of human melanoma-initiating cells[11,37] and participates in dormancy regulation of melanoma tumor-repopulating cells[38]. SOX2 has been recently related to the acquisition of an aggressive oxidative tumor phenotype endowed with enhanced drug resistance and metastatic ability[39]. Besides SOX2, the GLI transcription factors GLI1 and GLI2 have been suggested to be involved in melanoma progression and metastasis[40–42] and to be responsible for maintenance of melanoma-initiating cells[14]. Our data indicate that the transcriptional regulation of ST3GAL1 by SOX2 is direct, whereas GLI1 acts both directly and indirectly via SOX2. This result is consistent with a previous report demonstrating that SOX2 is a target of GLI1 in melanoma[11]. In addition, our results suggest the existence of reciprocal transcriptional regulation between SOX2 and GLI1. This regulatory loop might further contribute to potentiate the transcriptional activation of ST3GAL1. Our in vitro data support the biological relevance of the regulation of ST3GAL1 by SOX2 and GLI1. Indeed, ST3GAL1 appears to mediate the effects of SOX2 and GLI1 on melanoma cell invasion. Furthermore, ectopic expression of ST3GAL1 rescues the effect of SOX2 and/or GLI1 depletion on melanoma cell invasiveness. Interestingly, other members of the O-glycan biosynthesis pathway appear to be regulated by both SOX2 and GLI1, suggesting

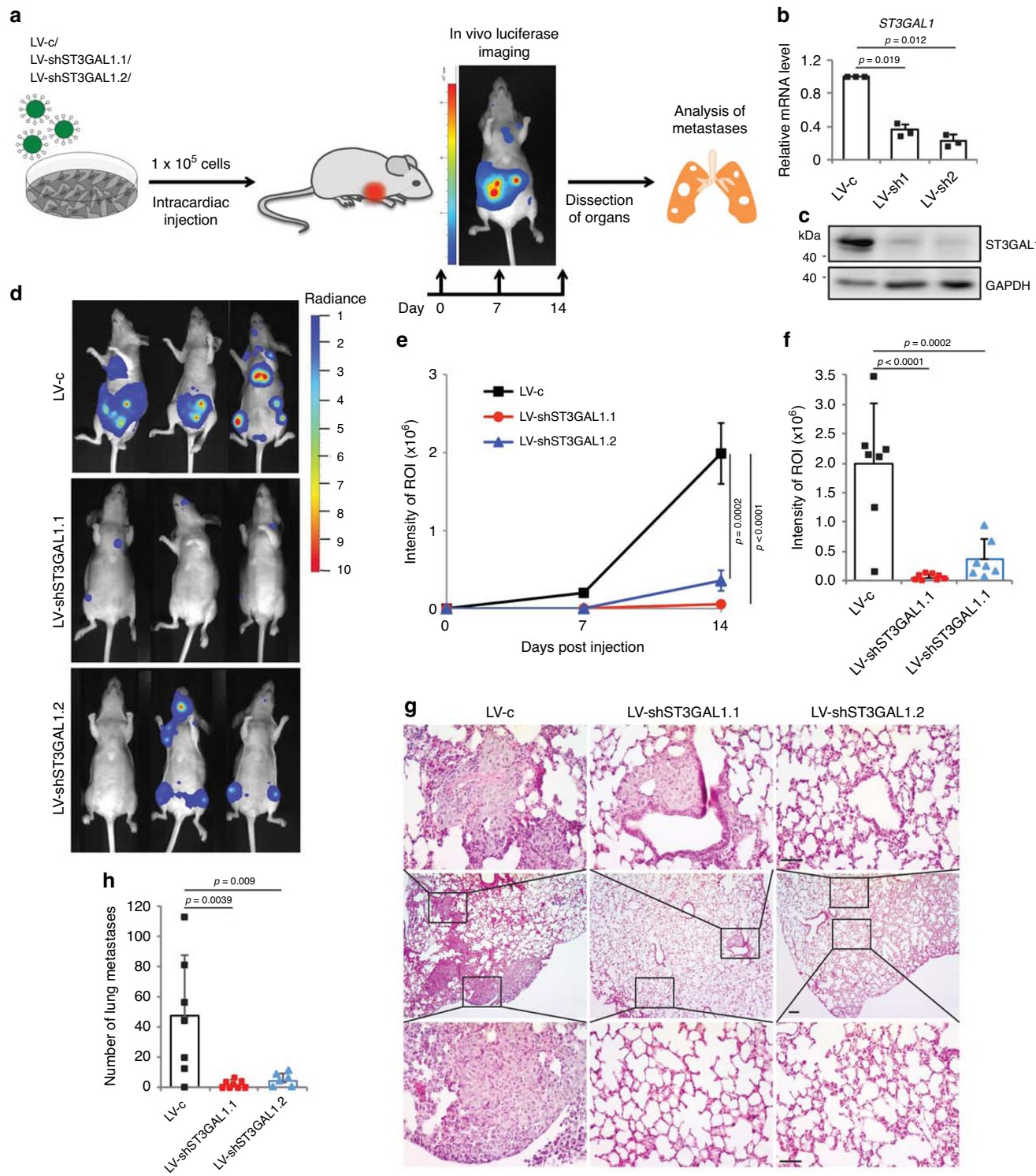

that these two TFs may promote melanoma progression through the induction of aberrant glycosylation.

Sialylation is a post-translational modification found in a large variety of proteins, including cell surface receptors and adhesion molecules. Sialylation is responsible for increased protein stability and activity, and proteins assembly. MS analysis in a melanoma cell line derived from a metastasis (A375 M6) identified several putative ST3GAL1 substrates, with enrichment in proteins involved in cell migration, locomotion, motility, and adhesion, all processes relevant for tumor metastasis. These proteins included the tyrosine kinase receptors AXL, NGFR, and EGFR, integrin β4

and β-catenin. However, only AXL is strongly activated in its catalytic domain by ST3GAL1[43]. In agreement with this finding, our data indicate that AXL is the main downstream mediator of ST3GAL1 function in a subgroup of human melanomas, given that genetic silencing of AXL rescues the increase in invasion induced by ectopic expression of ST3GAL1. Consistently, our data rule out the involvement of EGFR in mediating ST3GAL1-dependent cell migration and invasiveness and narrow that of NGFR in ST3GAL1-dependent melanoma cell migration. However, we cannot exclude the possibility that other RTK or proteins can play a role in mediating ST3GAL1 functions.

**Fig. 4 ST3GAL1 silencing decreases seeding of melanoma cells in the metastatic environment in vivo. a** Schematic representation of the in vivo metastasis assay. A375 M6 melanoma cells were transduced with LV-c, LV-shST3GAL1.1, or LV-shST3GAL1.2 and injected into the left ventricle of the heart of athymic nude mice ($n = 8$ per group). **b** qPCR of ST3GAL1 in A375 M6 transduced with control (LV-c), LV-shST3GAL1.1 or LV-shST3GAL1.2. Data are expressed as mean ± s.e.m. P value was calculated by two-tailed unpaired Student's $t$ test ($n = 3$ biological independent samples). **c** Representative western blot of ST3GAL1 in A375 M6 injected in mice. GAPDH was used as loading control. **d** Representative images of whole body in vivo BLI of mice injected with A375 M6 transduced as indicated at day 14 post-injection. **e** Intensity of ROI measured by in vivo bioluminescence imaging (BLI) at 7 and 14 days post-injection. Data are represented as mean ± s.e.m. P value was calculated by ANOVA and Tukey's test (LV-c $n = 7$, LV-shST3GAL1.1 $n = 8$, and LV-shST3GAL1.2 $n = 7$). **f** Quantification of BLI at day 14. Data are expressed as mean ± s.d. P value was calculated by ANOVA and Tukey's test (LV-c $n = 7$, LV-shST3GAL1.1 $n = 8$, LV-shST3GAL1.2 $n = 7$). **g** Representative bright-field microscopic images of lung metastases at the end of the experiment stained with haematoxylin and eosin (LV-c $n = 7$, LV-shST3GAL1.1 $n = 8$, and LV-shST3GAL1.2 $n = 6$) as shown in **h**. Scale bars 150 μm. **h** Quantification of lung metastases. Data are represented as mean ± s.d. P value was calculated by ANOVA and Tukey's test (LV-c $n = 7$, LV-shST3GAL1.1 $n = 8$, and LV-shST3GAL1.2 $n = 6$). Source data are provided as Source Data file.

AXL is a member of the TAM family of receptor tyrosine kinases. AXL is specifically activated by the binding of its ligand Gas6, which induces AXL homodimerization and subsequent autophosphorylation of multiple tyrosine residues in the cytoplasmic kinase domain[44]. Our data suggest that ST3GAL1 is required for ligand-dependent AXL dimerization and consequent autophosphorylation at residue Tyr702. Thus, we hypothesize that sialylation might be responsible for increasing the affinity of AXL for its ligand Gas6. Indeed, several glycosylation sites have been identified in the two Ig-like domains for ligand binding[45]. Clinically, the expression of AXL has been shown to correlate with a poor prognosis and to increase metastatic risk in several tumors, where it is implicated in cancer proliferation, survival, apoptotic evasion, and invasion, as well as in the acquisition of a stem-cell phenotype and in drug resistance[25]. Accumulating evidence indicate that AXL is a marker of invasive phenotype in melanoma, linked to the absence of microphthalmia-associated transcription factor (MITF) and the expression of an EMT signature[46,47]. Moreover, AXL expression has been also related with resistance to MAPK pathway inhibitors[32,46,48], and has been reported to desensitize melanoma cells to chemotherapy by preventing p53 activation[49]. However, AXL has been shown to have a dual regulatory function on melanoma cell invasion, given that both inhibition or overexpression enhance invadopodia formation and activity, due to compensatory mechanisms by ERBB3 signaling pathway[31].

Taken together, our study reveals a functional SOX2/GLI1-ST3GAL1-AXL axis in a subgroup of melanomas involved in cancer progression, and highlights the therapeutic potential of targeting ST3GAL1 to prevent or treat metastatic melanoma. The biosynthetic machinery involved in aberrant glycosylation has the potential to become a promising target for the development of anti-cancer drugs[50]. However, very few of these agents have the features required to translate into clinical trials. Our work provides the rationale for designing small molecules against ST3GAL1 and for drug repurposing to identify effective ST3GAL1 inhibitors for treatment of a subset of melanomas.

## Methods

**Cell cultures**. Normal human epidermal melanocytes (NHEM) were purchased from PromoCell (Heidelberg, Germany). Human melanoma cell lines A375, SK-Mel-2, SK-Mel-5, SK-Mel-28, MeWo, and HEK-293T were obtained from ATCC, whereas 501-Mel, A2058 and SK-Mel-197 were kindly provided by Dr. Laura Poliseno (CNR, Pisa, Italy). A375 M6 cells were isolated from lung metastases after tail-vein injection in SCID bg/bg mice of A375 cells[20]. Patient-derived SSM2c and M51 cells were obtained from metastatic melanomas[14,51] (Supplementary Table 3). All cells were cultured in Dulbecco's modified Eagle's medium (DMEM; Euroclone, Milan, Italy) containing 10% fetal bovine serum (FBS), 1% penicillin/streptomycin (PS), and 1% Glutamine, and maintained at 37 °C in a 5% $CO_2$ incubator. Short-term (passage 1–2) patient-derived melanoma cells Me-14, Me-16, Me-17, Me-25, Me-28, Me-32, Me-41, Me-42, and Me-44 were obtained from patients with primary or metastatic melanomas. Fresh tissue samples were enzymatically digested

with 20 μg/ml DNase I (Roche Applied Science, Basel, Switzerland) and 1 mg/ml collagenase A in Dulbecco's modified Eagle's medium (DMEM)-F12 (Euroclone, Milan, Italy) and filtered. Cells were grown in DMEM-F12 supplemented with 10% FBS, 2% PS, 1% Glutamine, and 5 ng/ml epidermal growth factor (EGF; Life Technologies, Paisley, UK). All cells were recently authenticated by DNA fingerprinting analysis and regularly tested by PCR to exclude Mycoplasma contamination. The use of melanoma samples was approved by the Ethic Committee of the University Hospital of Careggi with protocol numbers N. BIO.13.009 and N. BIO.14.026. All the subjects gave their written informed consent to participate in the study.

**Plasmids and viral production**. Lentiviruses for gene knockdown were produced in HEK-293T cells by cotransfecting lentiviral vector, dR8.74 packaging plasmid (Addgene #22036) and pMD2.G envelope plasmid (Addgene #12259). shRNA vectors used were: pLKO.1-puro (scramble, LV-c; Addgene #8453), pLKO.1-puro-shGLI1 (targeting sequence 5′-CCTGATTATCTTCCTTCAGAA-3′), pLKO.1-puro-shSOX2.1 (targeting sequence 5′-CTGCCGAGAATCCATGTATAT-3′), pLKO.1-puro-ST3GAL1.1 (LV-shST3GAL1.1) targeting the 3′ untranslated region (3′-UTR) of ST3GAL1 (targeting sequence 5′-AGAGACTTGAGTGGCGATTAC-3′) and pLKO.1-puro-shST3GAL1.2 (LV-shST3GAL1.2) targeting the coding region (CDS) of ST3GAL1 (targeting sequence 5′-GATGCAGACTTTGAGTCTAAC-3′), pLKO.1-puro-AXL.1 (LV-AXL.1) targeting the 3′-UTR of AXL (targeting sequence 5′- CTTT AGGTTCTTTGCTGCATT-3′) and pLKO.1-puro-shAXL.2 (LV-shAXL.2) targeting the CDS of AXL (targeting sequence 5′-GATTGCCATTGAGAGTCTAGC-3′); pLKO.1-puro-NGFR.1 (LV-shNGFR.1) targeting the 3′-UTR (targeting sequence 5′-GCACTGTAGTAAATGGCAATT-3′) and pLKO.1-puro-NGFR.2 (LV-shNGFR.2) targeting the CDS of NGFR (targeting sequence 5′-CCTCCAGAACAAGAC CTCATA-3′); pLKO.1-puro-shEGFR.1 (LV-shEGFR.1) targeting the 3′-UTR (targeting sequence 5′-GAGAATGTGGAATACCTAAGG-3′) and pLKO.1-puro-shEGFR.2 (LV-shEGFR.2) targeting the CDS of EGFR (targeting sequence 5′-GCCA-CAAAGCAGTGAATTTAT-3′). When unspecified, gene silencing was performed by co-infection of melanoma cells with both CDS- and 3′-UTR-targeting lentiviruses.

Lentiviruses for gene overexpression were produced in HEK-293T by co-transfection of CSGW vector, CSGW-SOX2[52] (cloned into the BglII-NotI restriction sites of CSGW vector using the following primers: SOX2-F 5′-ATGTACAACATGAT GGAGACGG-3′ and SOX2-R 5′- TCACATGTGTGAGAGGGGC-3′) or CSGW-GLI1 (cloned into the BglII-NotI restriction sites of CSGW vector using the following primers: GLI1-F 5′- ATGTTCAACTCGATGACCCCAC-3′ and GLI1-R 5′-TTAGG CACTAGAGTTGAGGAA-3′), pCMV-dR8.91 packaging plasmid (gift from Silvestro Conticello) and pCMV-VSV-G envelope plasmid (Addgene #8454). Retroviruses for gene overexpression were also produced in HEK-293T by cotransfecting pBABE-puro (Addgene #1764), pBABE-ST3GAL1 (cloned into the BamHI/SnaBI restriction sites of pBABE-puro vector using the following primers: ST3GAL1-F 5′-ATGGTGACCC TGCGGAAGAGG-3′ and ST3GAL1-R 5′-TCATCTCCCCTTGAAGATCCGGA-3′) or pWZL-Neo-Myr-Flag-AXL (Addgene #20428) with pUMVC packaging plasmid (Addgene #8449) and pCMV-VSV-G envelope plasmid (Addgene #8454).

**RNA extraction and real-time qPCR**. Total RNA was extracted with RNeasy mini kit (Qiagen) and quantified by using Nanodrop 8000. After DNase I treatment (Roche Diagnostics), 1 μg of RNA was reverse transcribed with High-Capacity RNA-to-cDNA™ Kit (Applied Biosystems) according to manufacturer's instructions. Quantitative real-time PCR was carried out at 60 °C using FastStart SYBR Green Master (Roche Diagnostics) in a Rotorgene-Q (Qiagen). Primers were designed by using Primer3. Primer sequences are reported in Supplementary Table 4.

**RNA sequencing**. For the RNA-seq analysis cDNA libraries were prepared following Illumina RNA-Seq preparation protocol. The cDNA fragments were amplified by polymerase chain reaction (PCR) and sequenced at both ends using an Illumina Genome Analyzer IIx. RNA-seq data were analyzed using MAP-RSEQ

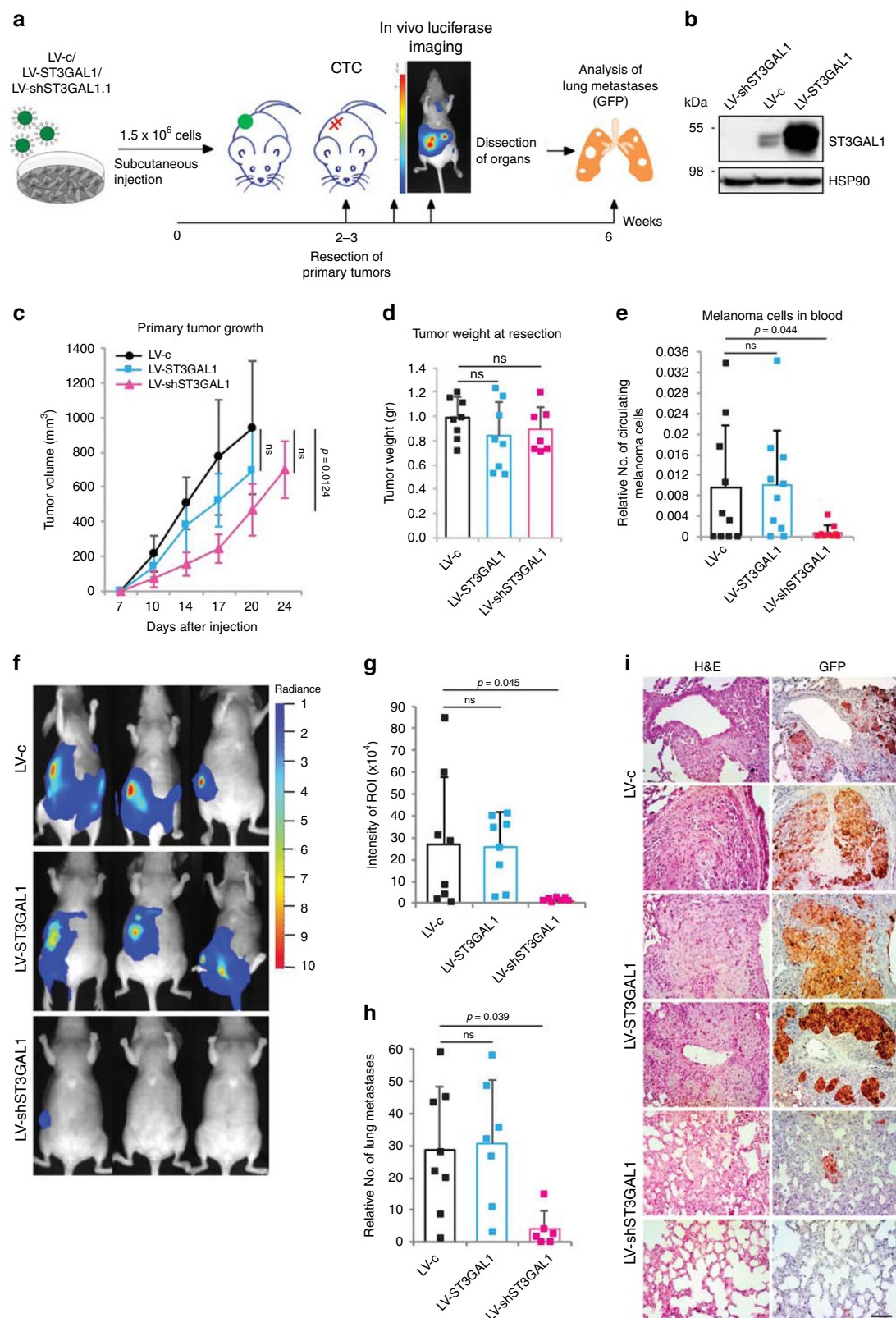

(http://bioinformaticstools.mayo.edu/), a high-throughput package designed by the Mayo Clinic Bioinformatics Program for these purposes.

**Patient-derived xenograft cell cultures and scRNA-seq.** Four cell types were used for single-cell RNA sequencing. Normal human neonatal epidermal mela-nocytes were purchased from Lifeline Cell Technology (FC-0019). These were briefly propagated in culture using the DermaLife M Melanocyte Medium with complete supplements (Lifeline Cell Technology, FC-0027). Melanoma patient-derived xenograft lines M12, M15, and M27 were grown in DMEM with 10% FBS. All PDXs were derived from the brain metastases of Mayo Clinic patients. These cells were harvested and sent to Mayo Clinic's Genome Analysis Core Facility where the cells were prepared for single-cell RNA sequencing. Tissue acquisition for M12, M15, and M27 melanoma brain metastases was approved by IRB and PDXs development by IACUC. All the subjects gave their written informed consent to participate in the study.

**Fig. 5 ST3GAL1 silencing decreases in vivo melanoma metastasis. a** Schematic representation of the in vivo xenograft experiment. A375 M6 melanoma cells were transduced with LV-c, LV-shST3GAL1.1, or LV-ST3GAL1 and injected subcutaneously into the flank of athymic nude mice ($n = 8$ per group). **b** Representative western blot of ST3GAL1 in A375 M6 cells used for injection in mice. HSP90 was used as loading control. **c** Primary tumor growth of A375 M6 cells transduced with LV-c, LV-shST3GAL1.1, or LV-ST3GAL1. Data are represented as mean ± s.d ($p = 0.0124$, day 20 LV-c vs LV-shST3GAL1; LV-c $n = 7$, LV-shST3GAL1 $n = 7$, LV-ST3GAL1 $n = 8$). **d** Tumor weight at the time of resection of the primary tumor (LV-c $n = 8$, LV-ST3GAL1 $n = 8$, LV-shST3GAL1 $n = 7$). **e** Relative number of GFP + circulating melanoma cells in the blood ($n = 8$ independent animals). Data are represented as mean ± s.d. $P$ value was calculated by two-tailed unpaired Student's $t$ test; ns not significant. **f** Representative images of whole body in vivo BLI of mice injected with A375 M6 transduced as indicated at day 45 post-injection (three weeks post-resection of lateral tumors). **g** Intensity of ROI measured by in vivo bioluminescence imaging (BLI) at 45 days post-injection (LV-c $n = 8$, LV-ST3GAL1 $n = 8$, and LV-shST3GAL1 $n = 7$). **h** Quantification of lung metastases (LV-c $n = 8$, LV-ST3GAL1 $n = 7$, and LV-shST3GAL1 $n = 6$). Data in panels **c**, **d**, **g**, and **h** are represented as mean ± s.d and $P$ value was calculated by ANOVA and Tukey's test (**c**, **d**, **h**) or ANOVA and Dunnett's test (**g**); ns not significant. **i** Representative bright-field microscopic images of lung metastases from lung serial sections stained with haematoxylin and eosin (left panels) or immunostained with GFP (right panels) (LV-c $n = 8$, LV-ST3GAL1 $n = 7$, and LV-shST3GAL1 $n = 6$) as shown in **h**. Scale bar 150 μm. Source data are provided as Source Data file.

For scRNA-seq cells were dissociated to single cells using a combination of centrifugation and straining, and resuspended to desired concentration based on cell type and sequencing goal. Whole live cells were washed twice in 1X PBS + 0.04% BSA and immediately single cell sorted. Cells were first counted and measured for viability using either the Vi-Cell XR Cell Viability Analyzer (Beckman-Coulter) or a basic hemocytometer and light microscopy. Based on the desired number of cells to be captured for each sample, a volume of live cells was mixed with the cDNA master mix. The cell suspension master mix, Gel Beads and partitioning oil were added to a Chromium Single Cell B chip. The filled chip was loaded into the Chromium Controller, where each sample was processed and the individual cells within the sample were captured into uniquely labeled Gel Beads-In-Emulsion (GEMs). The GEMs were collected from the chip and taken to the bench for reverse transcription, GEM dissolution, and cDNA clean-up. The resulting cDNA contained a pool of uniquely barcoded molecules. A portion of the cleaned and measured pooled cDNA continued on to library construction, where standard Illumina sequencing primers and a unique i7 Sample index were added to each cDNA pool. All cDNA pools and resulting libraries were measured using Qubit High Sensitivity assays (Thermo Fisher Scientific), Agilent Bioanalyzer High Sensitivity chips (Agilent) and Kapa DNA Quantification reagents (Kapa Biosystems). Libraries were sequenced at 60,000 fragment reads per cell following Illumina's standard protocol using the Illumina cBot and HiSeq 3000/4000 PE Cluster Kit. The flow cells were sequenced as 100 × 2 paired end reads on an Illumina HiSeq 4000 using HiSeq 3000/4000 sequencing kit and HCS v3.3.52 collection software. Base-calling was performed using Illumina's RTA version 2.7.3.

The single-cell RNA-seq dataset was analyzed using Monocle3[53]. The gene expression matrices of all 4 libraries were combined and genes expressed in <3 cells were filtered out. Cells with the number of genes detected <500, or total UMI fewer than 1000 or more than 102,784 (three standard deviations above the mean), or having mitochondrial content >50% were discarded. After these initial filtering steps the gene counts were log-transformed after adding a pseudocount, and each gene was normalized to a Gaussian distribution with mean of 0 and variance of 1. The dimensionality of the data was reduced by principal component analysis (PCA) and only the first 100 components were used for downstream analysis. For visualization and clustering purpose the data were further projected into two-dimensional space using the UMAP algorithm [https://arxiv.org/abs/1802.03426v2]. Cells were then clustered based on the UMAP projection using the Leiden community detection algorithm [https://www.nature.com/articles/s41598-019-41695-z]. To determine marker genes unique to each cluster, we used the top_markers function of Monocle3, where 1000 cells from each cluster were randomly selected as reference set for marker significance testing.

**Microarray analyses.** Publicly available gene expression data were: GSE46517 (31 primary and 73 metastatic melanomas); GSE7553 (4 normal skin, 11 squamous cell carcinoma, 15 basal cell carcinoma, 2 melanoma in situ and 54 malignant melanomas); GDS1375 (17 nevi and 45 malignant melanomas); GDS3966 (31 primary and 52 metastatic melanomas). All data sets were profiled on Affymetrix U133 platform.

**In vitro invasion and wound-healing assays.** Cell invasion was performed using 24-well Corning transwell membranes pre-coated with Matrigel (0.4 mg/ml; Becton Dickinson). Briefly, A357 M6 (50,000 cells/well), SSM2c, SK-Mel-28, MeWo, and M51 (100,000 cells/well) were suspended in serum-free medium supplemented with 500 μg/ml Mitomycin C (Sigma-Aldrich), and seeded over the Matrigel coating. Medium supplemented with 20% FBS was used as a chemo-attractant. After 30 h invaded cells were stained with Diff Quick (MediCult Italia S.p.A) and counted.

For would-healing, 30,000 melanoma cells were seeded on a plate using Culture 2 well silicone inserts (IBIDI) and let to adhere overnight in complete medium until they reached a confluence of ~90%. The inserts were then removed, medium replaced by serum-free DMEM added with 0.5 mg/ml mitomycin C, and the

quality of the covered area was evaluated. The cell-free scratches were imaged at indicated different time points after insert removal, using a LEICA DFC450C microscope with ×4 objective lens, until complete wound closure. The measure of cell-free scratch was performed with Image J software and the relative migration rate calculated as the mean of the relative percentage of covered area compared to the T0 area for each well.

**Intracardiac metastasis model in vivo.** To assess the effect of ST3GAL1 depletion on melanoma cell seeding and survival in a metastatic environment, we used a model of intracardiac instillation of melanoma cells. A375 M6 cells were first transduced with LV-c, LV-shST3GAL1.1, or LV-shST3GAL1.2, and then stably transfected with a luciferase reporter plasmid (pGL4.51[luc2/CMV/Neo] Vector). Cells were resuspended in DMEM at the concentration of $5 \times 10^5$ cells per 150 μl, aliquoted into Eppendorf tubes and maintained on ice until injection. On day 0, female athymic nude 8-week-old mice were anesthetized by exposure to isoflurane and injected into the left ventricle of the heart with $5 \times 10^5$ A375 M6 cells transduced as indicated above. A successful intracardiac injection was indicated on day 1 by images showing systemic bioluminescence distributed throughout the animal. Only mice with evidence of a satisfactory injection were kept in the experiment. Assessment of metastases was performed weekly by measuring bioluminescence produced by metastatic cells in the living mice. Briefly, the substrate luciferin was injected into the intraperitoneal cavity at a dose of 150 mg/kg body weight (30 mg/ml), ~15 min before imaging. Mice were anesthetized with 2.5% Avertin and placed on the imaging stage. Ventral images were collected for automatic exposure (3 min) using the Photon Imager system (Biospace Lab). Analysis was performed using M3Vision software (Biospace Lab) by measurement of photon flux (measured in photons/s/cm2/steradian) with a region of interest (ROI) drawn around the bioluminescence signal to be measured. Data were plotted using GraphPad PRISM and significance was determined by unpaired $t$ test. After killing mice, metastasis-bearing lungs and other organs were dissected, fixed in 10% formalin and processed for histology.

**In vivo metastasis assay.** To assess the role of ST3GAL1 in the formation of metastases we used a xenograft model. A375 M6 cells were first stably transfected with a luciferase reporter plasmid (pGL4.51[luc2/CMV/Neo] Vector), and then co-transduced with LV-GFP and either LV-c, LV-shST3GAL1.1, or LV-ST3GAL1. Cells were resuspended in DMEM at the concentration of $1.5 \times 10^6$ cells per 150 μl, aliquoted into Eppendorf tubes and maintained on ice until injection. On day 0, 8-week-old female athymic nude mice were anesthetized by exposure to isoflurane and injected subcutaneously into the right flank ($n = 8$ for each group).

When primary tumors were palpable, measurements were made with a caliper twice a week until resection. Tumor volume was calculated with the formula $V = W^2 \times L \times 0.5$, where $W$ represents tumor width and $L$ the length. Primary tumors were surgically resected when tumor volume reached 800 mm³. Assessment of metastases was performed weekly by measuring bioluminescence produced by metastatic cells in the living mice by injecting the substrate luciferin, as described above. After killing mice, metastasis-bearing lungs and other organs were dissected, fixed in 10% formalin and processed for histology. All animal protocols were approved by local ethic authorities (CeSAL, Centro Stabulazione Animali da Laboratorio) and by the Italian Ministry of Health and conducted in accordance with Italian Governing Law (D.lgs 26/2014). In all in vivo experiments, mice were maintained at the animal facility (CeSAL) of the University of Florence (Italy). Animals were maintained in a pathogen-free, temperature-controlled, 12 h light and dark cycle environment, and were fed ad libitum. Mice were housed in plastics cages (no more than four animals per cage to minimize aggressiveness).

**Flow cytometry.** For apoptosis analysis, cells were serum-starved for 48 h and apoptosis was measured using the Annexin V–phycoerythrin/7-AAD apoptosis kit (BD Biosciences, San Diego, CA, USA) according to the manufacturer's protocol. The number of Annexin V$^+$/7-AAD$^-$ (early apoptosis) and Annexin V$^+$/7-AAD$^+$

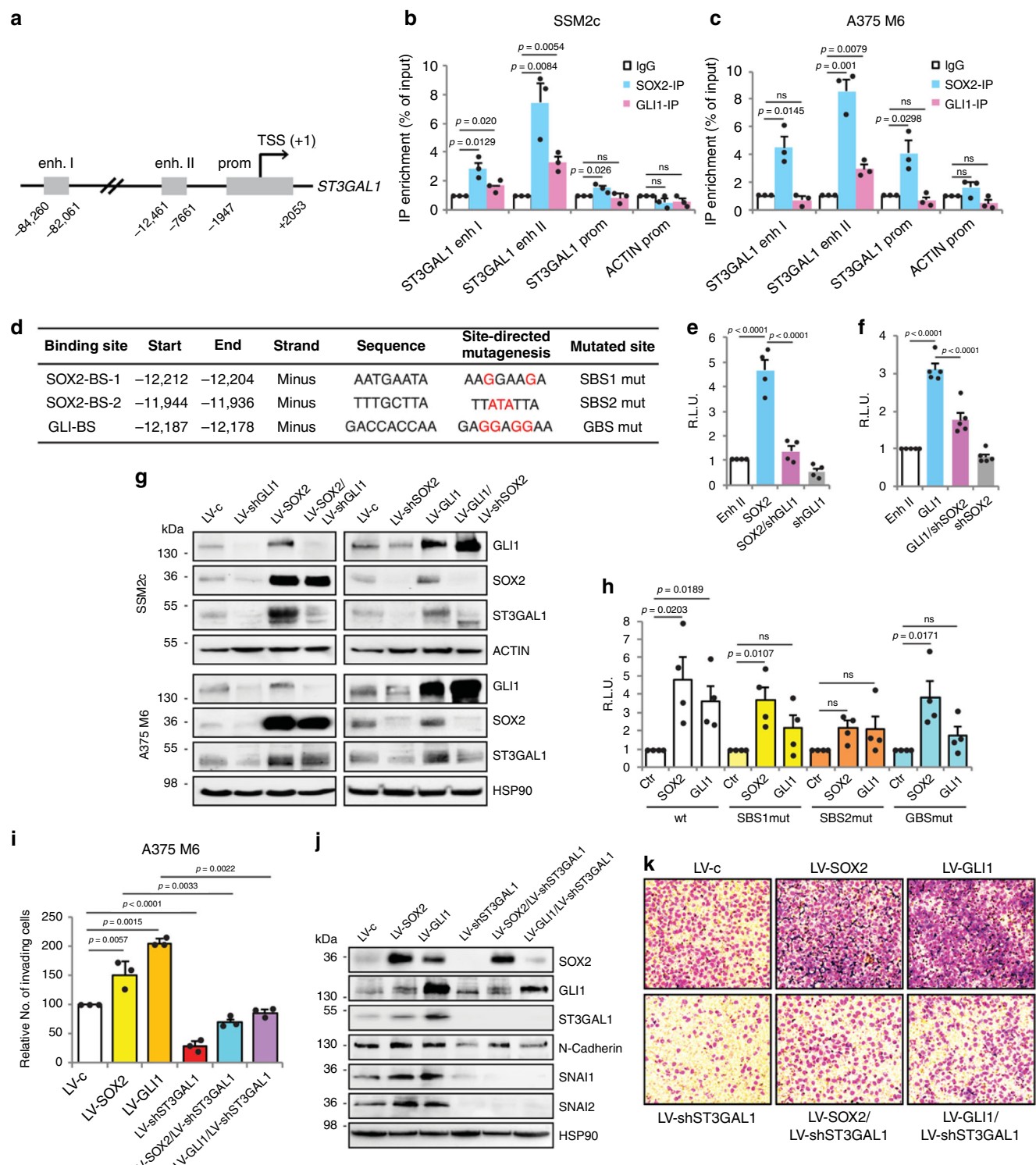

(late apoptosis) labeled cells was detected with a CytoFLEX S Flow Cytometer (Beckman Coulter) and analyzed using the CytExpert Software.

For circulating melanoma cells analysis, 100 µl of blood was collected from mice at surgery of the primary tumors by cardiac puncture through a syringe using heparin as anticoagulant[54]. Red blood cells (RBC) were removed by addition of RBC lysis solution (83 mg $NH_4Cl$, 10 mg $KHCO_3$, 18 µl EDTA 5%, $ddH_2O$ to a final volume of 10 ml) followed by centrifugation at $1200 \times g$ for 10 min. Pellets were resuspended in 300 µl in PBS before flow cytometry. GFP⁺-cells were detected with a CytoFLEX S Flow Cytometer. Sorting gates were drawn using A375 M6 GFP⁻ and GFP⁺ cells as the negative or the positive control, respectively. Blood sample from healthy mice was also used to exclude unspecific fluorescence signals from murine blood cells. Data were analyzed with FlowJo software (TreeStar Inc., Ashland, OR, United States) and normalized on number of events.

All FACS sorting gate strategies are provided in Supplementary Fig. 13.

**Immunohistochemistry and histopathological analysis**. Immunohistochemistry was performed on tissue microarrays (US Biomax, Inc.) of formalin-fixed paraffin-embedded specimens of human nevi ($n = 24$), malignant melanomas ($n = 56$), and metastatic malignant melanomas ($n = 40$). After antigen retrieval (with citrate buffer pH 6.0), staining was performed with the UltraVision Detection System kit (Lab Vision, Fremont, CA, USA) following manufacturer's instructions. Sections

**Fig. 6 SOX2 and GLI1 co-regulate *ST3GAL1* gene transcription. a** Schematic representation of *ST3GAL1* regulatory regions (RRs). **b, c** ChIP-qPCR of SOX2 and GLI1 occupancy of the ST3GAL1 RRs showing that both TFs bind to *ST3GAL1* enhancer II in SSM2c (**b**) and A375 M6 (**c**) cells. The *y*-axis represents relative promoter enrichment, normalized on input material. IgG was set to 1. *ACTIN* promoter was used as negative control. Data are represented as mean ± s.e.m. *P* value was calculated by two-tailed unpaired Student's *t* test (*n* = 3 biological independent experiments); ns not significant. **d** Consensus SOX2 and GLI1 binding sites identified in *ST3GAL1* enhancer II. **e, f** Quantification of dual-luciferase reporter assay in SSM2c melanoma cells showing that silencing of GLI1 or SOX2 is able to counteract the transactivation induced by SOX2 (**e**) or GLI1 (**f**) on the enhancer II of *ST3GAL1*. Relative luciferase activities were firefly/Renilla ratios, with level induced by control equated to 1. Data are represented as mean ± s.e.m. *P* value was calculated using ANOVA and Tukey's test (*n* = 4 in **e**, *n* = 5 in **f**). **g** Western blot of SOX2, GLI1, and ST3GAL1 in A375 M6 and SSM2c cells transduced as indicated. **h** Quantification of dual-luciferase reporter assay in SSM2c cells showing that mutagenesis of GLI1-BS and SOX2-BS-2 prevents SOX2 and GLI1 from transactivating *ST3GAL1* enhancer II. Relative luciferase activities were firefly/Renilla ratios, with the level induced by the control equated to 1. Data are represented as mean ± s.e.m. by two-tailed unpaired Student's *t* test (*n* = 4). **i** Matrigel invasion assay in A375 M6 cells transduced as indicated. Data are represented as mean ± s.d. using two-sided Kruskal–Wallis and Dwass-Steel-Critchlow-Fligner method (*n* = 3 biological independent experiments). **j** Western blot of SOX2, GLI1, ST3GAL1, and EMT markers in A375 M6 cells transduced as indicated. **k** Representative images of **i**. Blots in **g** and **j** are representative of *n* = 3 biological independent experiments. HSP90 was used as loading control. Source data are provided as Source Data file.

were incubated overnight at 4 °C with rabbit anti-ST3GAL1 antibody (1:50 dilution; #PA5-21721, Invitrogen). AEC (3-amino-9-ethylcarbazole; Dako, Copenhagen, Denmark) was used as chromogen. Sections were counterstained with hematoxylin. Each stained core section was scored by visual microscopy inspection as follows: 0, for no staining (negative); 1, for weak staining (low); 2, for moderate staining (medium); and 3, for marked staining (high). Most of the cores showed expression in >75% of the tumor cells. Immunohistochemistry in formalin-fixed paraffin-embedded murine lung sections was performed as reported above, using anti-GFP antibody (1:500 dilution; Santa Cruz Biotechnology).

**Chromatin Immunoprecipitation.** For ChIP experiments, $3 \times 10^6$ SSM2c cells were fixed with 1% formaldehyde for 15 min, and fixation stopped by adding 125 mM glycine for 5 min. Cells were lysed in Farnham lysis buffer (5 mM PIPES pH 8, 85 mM KCl, 0.5% NP-40) added with protease inhibitors for 10 min. Nuclei were collected by centrifugation at $800 \times g$ for 10 min and then lysed in nuclear lysis buffer (1% SDS, 10 mM EDTA, 50 mM Tris-HCl pH 8) added with protease inhibitors. Chromatin was sonicated to an average size of 200–600 bp with a SONOPULS Mini20 Sonicator (Bandelin), diluted with ChIP Dilution Buffer (10 mM Tris-HCl pH 8, 2 mM EDTA, 140 mM NaCl, 1% Triton X-100, 0.1% SDS) and incubated overnight with 20 µl protein G magnetic dynabeads and 3 µg of mouse anti-SOX2 (R&D System, #MAB2018) (1:1000), mouse anti-GLI1 (Cell Signaling, #2643) (1:500), or normal mouse IgG (Santa Cruz Biotechnology, #sc-2025) (1:100) antibodies. Immunocomplexes were washed with increasing salt concentrations, DNA was eluted at 85 °C with 1% SDS and crosslinks reversed overnight at 65 °C with 200 mM NaCl. DNA was treated with 4 µg RNase A (Thermo Fisher Scientific) and 20 µg Proteinase K (Qiagen) at 37 °C for 2 h to remove RNA and protein contaminations, and recovered with QIAquick PCR Purification Kit (Qiagen). qPCR was carried out at 60 °C using FastStart SYBR Green Master (Roche) in a Rotorgene-Q (Qiagen). Primer sequences are reported in Supplementary Table 4.

**Luciferase reporter assay and mutagenesis.** A fragment of ST3GAL1 enhancer II (−12461/−11451bp from TSS) was PCR amplified with KOD hot start DNA polymerase (Merck Millipore) and cloned into the pGL3Basic vector (Promega) using NheI-HindIII sites, to generate ST3GAL1 enh-luc reporter. Primers used were: ST3GAL1 enh F, 5′-AGTTGCCATCCCCTAAGGTAA-3′; ST3GAL1 enh R, 5′-AGGAACCATGATCCTTATGGA-3′. Mutations of ST3GAL1 enh-luc reporter were introduced using QuickChange II (Agilent Technologies) with the following oligos: SBSmut1 F, 5′-TTTGTCAAAAATCAGCTCATGAAGGAATAGCTTTAATCCCTACTTGG-3′; SBSmut1 R, 5′-CCAAGTAGGGATTAAAGCTATTCCTTCATGAGCTGATTTTTGACAAA-3′; SBSmut2 F, 5′-GTTCTGGACTTTCTGTTTCATGAGTCAATAAAATTCTTTATATTACTCCACTTTAGGTCATAG-3′; SBSmut2 R, 5′-CTATGACCTAAAGTGGAGTAATATAAAGAATTTTATTGACTCATGAAACAGAAAGTCCAGAAC-3′; GBSmut F, 5′-ATGAATGAATAGCTTTAATCCCTACTTGGAGGACCAAGATAAAGAGATAAAT-3′; GBSmut R, 5′-ATTTATCCTTTATCTTTGGTCCTCCAAGTAGGGATTAAAGCTATT-CATTCAT-3′. ST3GAL1 enh-luc reporters were used in combination with *Renilla* luciferase pRL-TK reporter vector (Promega) to normalize luciferase activities. The sequence of ST3GAL1 enhancer II is reported in Supplementary Fig. 4.

**Protein extraction and western blot.** Cells were lysed in cold RIPA buffer (50 mM Tris-HCl pH 7.5, 1% NP-40, 150 mM NaCl, 5 mM EDTA, 0.25% NaDOC, and 0.1% SDS) supplemented with protease and phosphatase inhibitors and centrifuged at $20,000 \times g$ for 20 min at 4 °C[55]. Supernatant was collected as whole cell extract (WCE). Equal amounts of protein were resolved by SDS-polyacrylamide gel electrophoresis, transferred onto nitrocellulose membranes, and incubated for 1 h in blocking buffer at room temperature. Primary antibodies are reported in Supplementary Table 5. Blotted membranes were developed by using SuperSignal West Femto (Thermo Fisher Scientific) and imaged with ChemiDoc Imaging Systems (Bio-Rad).

**Detection of sialylation by lectin affinity immunoprecipitation.** For immunoprecipitation (IP) of sialylated proteins with MAL-II lectin, cells were lysed in cold IP buffer (0.5% NP-40, 100 mM NaCl, 5 mM EDTA, 10% glycerol, and 50 mM Tris-HCl pH 7.5) supplemented with protease and phosphatase inhibitors, and supernatant was collected as WCE after centrifugation at $20,000 \times g$ for 15 min at 4 °C. In all, 1 mg WCE was diluted with IP buffer to a final volume of 500 µl and incubated for 3 h at room temperature with 10 µg of biotinylated Maackia Amurensis Lectin II (MAL-II; Vector Laboratories). Streptavidin-agarose beads (Thermo Fisher Scientific) were then added and samples were incubated for additional 2 h at room temperature in constant rotation. Lectin-bounded sialylated proteins were collected after brief centrifugation, washed with IP buffer, eluted from beads by boiling in SDS–PAGE sample buffer 2X and resolved by SDS-polyacrylamide gel electrophoresis.

**AXL activation and dimerization.** In all, 600,000 A375 M6 cells were seeded in 60 mm dishes in complete medium to allow them to adhere overnight, serum-starved for 16 h and stimulated for 1 hr with 250 ng/ml of human recombinant GAS6 (R&D Systems, 885-GSB-050) on ice. Cells were then crosslinked with 3 mM BS3 (Sigma-Aldrich, Cat.no. 82436-77-9) for 20 min on ice and quenched with 250 mM glycine for 5 min at 4 °C[56]. Pellets were lysed in ice-cold lysis buffer (140 mM NaCl, 10 mM EDTA, 10% glycerol, 1% NP-40, and 20 mM TRIS-HCl pH 8) supplemented with protease and phosphatase inhibitors, and supernatant was collected as WCE after centrifugation at $16,000 \times g$ at +4 °C for 20 min. In total, 50 µg proteins were resolved on a 7% SDS-polyacrylamide gel, transferred overnight onto a nitrocellulose membrane and incubated with anti-AXL antibody (Santa Cruz Biotechnology, sc-166269).

**Mass spectrometry analysis.** Purification of sialylated proteins for Mass Spectrometry (MS) analysis was performed with MAL spin columns in the Qproteome Sialic Glycoprotein Kit (Qiagen) according to the manufacturer instructions. In total, 40 µg of sialylated proteins enriched from LV-shST3GAL1 and from LV-ST3GAL1 were trypsin digested with a filter aided sample preparation (FASP) protocol[57]. After digestion, tryptic peptides were desalted using C18 cartridges (Sep-pack, Waters) according to manufacturer's instructions. Eluted peptides were dried under vacuum and suspended in 40 µL of CH$_3$CN 3%/formic acid 0.1%. In all, 1 µL of sample was used for each round of analysis. LC-MS/MS experiments were performed with a LTQ-Orbitrap XL mass spectrometer (Thermo Fisher Scientific) coupled online to a nano-HPLC Ultimate 3000 (Dionex – Thermo Fisher Scientific). Samples were loaded into a 10-cm pico-frit column (75 µm I.D., 15 µm tip, New Objective) packed in-house with C18 material (Aeris Peptide 3.6 lm XB-C18; Phenomenex). Each sample was repeatedly injected to obtain technical replicates of the experiment. Peptides were separated at a flow rate of 250 nL/min using a linear gradient of CH$_3$CN from 3% to 40% in 90 min. A data dependent acquisition (DDA) with a top4 method was used for the analysis: one full scan between 300 and 1700 *m/z* at high resolution (60,000) in the Orbitrap was followed by the CID fragmentation in the linear trap of the four most intense ions. Raw data files were analyzed with the software package MaxQuant and the search engine Andromeda[58,59], against the human section of the Uniprot database (version july 2018) concatenated with a database of contaminants commonly found in proteomics experiments. Trypsin was selected as enzyme with two missed cleavages allowed; carbamidomethylation of cysteine residues was set as fixed modification and methionine oxidation as variable modification. Results were filtered at a false discovery rate (FDR) of 0.01, both at the peptide and protein level. Only proteins identified with at least three independent peptides were considered as a significant hit. The parameter "LFQ" calculated by the software was used to estimate the abundance of the proteins across the different samples. A Student's *t* test was performed to highlight proteins with a significant ($p \leq 0.05$) different abundance in LV-shST3GAL1 and LV-ST3GAL1 samples. A fold change ≥1.5 was selected to classify proteins as potential substrates of ST3GAL1.

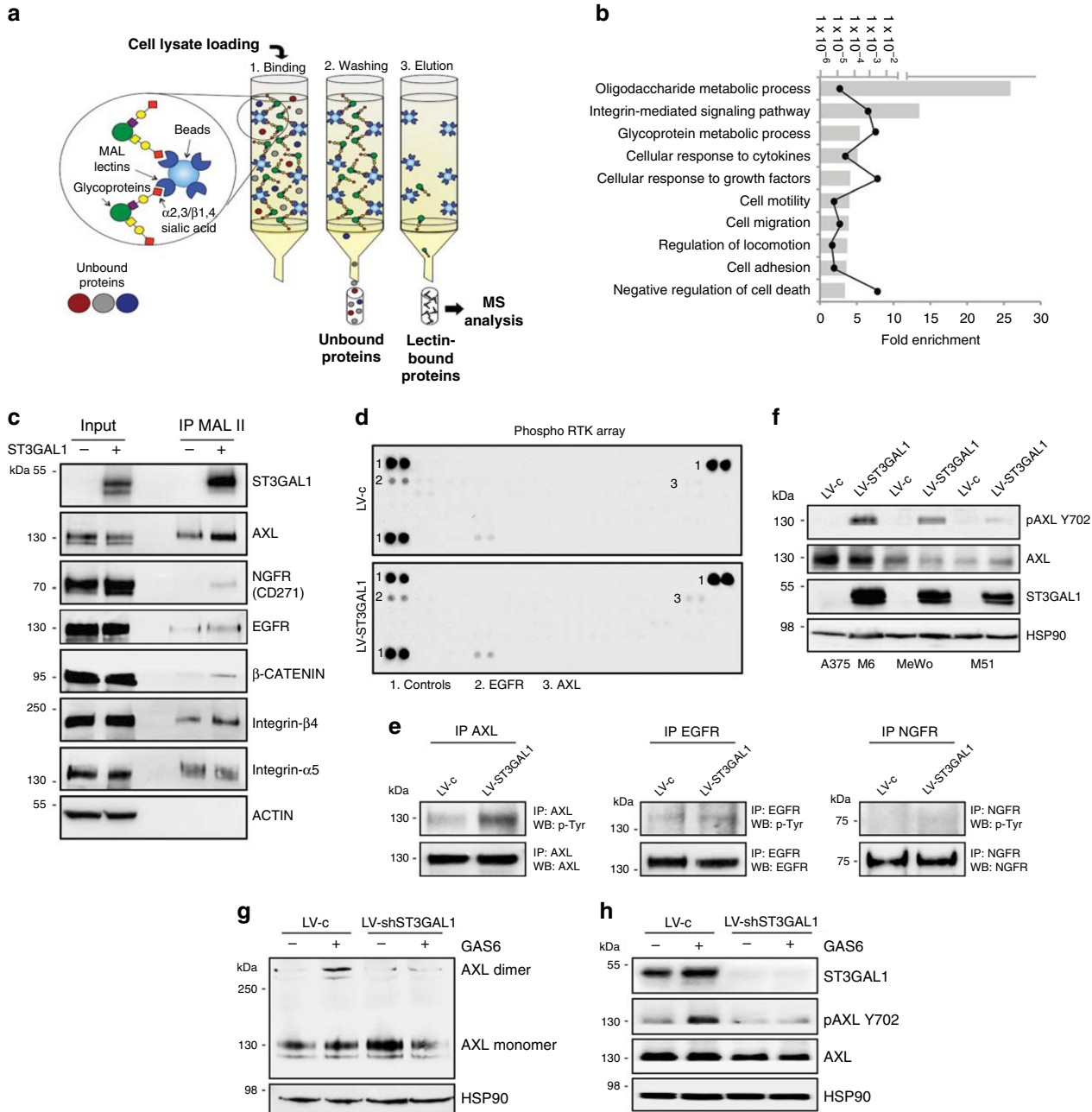

**Fig. 7 Identification of AXL as a ST3GAL1 substrate in melanoma cells. a** Schematic illustration of the experimental approach showing affinity enrichment of sialylated proteins by MAL lectin affinity chromatography in A375 M6 cells. **b** Gene ontology enrichment analysis (category of biological processes) of sialylated proteins in LV-ST3GAL1 versus LV-shST3GAL1.1 A375 M6 cells. Black line indicates *p* values. See also Supplementary Data 2 and Supplementary Table 2. **c** MAL-II lectin affinity immunoprecipitation (IP) of whole-cell lysates of A375 M6 transduced with LV-ST3GAL1 (+) or LV-shST3GAL1.1 (−), followed by western blot of ST3GAL1, AXL, NGFR, EGFR, β-Catenin, Integrin β4, or Integrin α5. ACTIN was used as loading control. Input was 5% of the total IP. **d** Representative phospho-receptor tyrosine kinase (RTK) assay in A375 M6 melanoma cells transduced with LV-c or LV-ST3GAL1. Numbers indicate loading controls[1] and activated RTKs[2,3] (*n* = 2 biological independent experiments). **e** Western blot of phospho-Tyrosine after immunoprecipitation with AXL, EGFR, or NGFR in A375 M6 cells transduced with LV-c or LV-ST3GAL1. **f** Western blot of pAXL-Y702, AXL, and ST3GAL1 in A375 M6, MeWo, and M51 cells transduced as indicated. HSP90 was used as loading control. **g** Dimerization of AXL in A375 M6 cells transduced with LV-c or LV-shST3GAL1, stimulated with Gas6 (250 ng/ml) for 1 h and crosslinked with BS3. Both AXL monomer and dimer were visible in the immunoblot. HSP90 was used as loading control. **h** Western blot of pAXL-Y702, AXL, and ST3GAL1 in A375 M6 transduced with LV-c or LV-shST3GAL1 upon Gas6 stimulation (250 ng/ml) for 1 h. HSP90 was used as loading control. Blots in **c** and **e**–**h** are representative of *n* = 3 biological independent experiments. Source data are provided as Source Data file.

**Phospho-receptor tyrosine kinase proteome array**. A Proteome Profiler Antibody Arrays Kit for human phospho-receptor tyrosine kinases (phospho-RTK, R&D System) was used to determine which RTK is phosphorylated in presence of ST3GAL1. Briefly, A375 M6 cells transduced with scramble (LV-c) or LV-ST3GAL1 were lysed for 30 minutes in cold lysis buffer (R&D System) supplemented with protease and phosphatase inhibitor cocktail (Sigma-Aldrich), and spun at 14,000 × *g* for 5 min to remove debris. Array membranes were blocked with the provided blocking buffer (R&D System) for 1 h, and then incubated overnight with the cell lysate at 4 °C on a rocking platform shaker. The day after, array membranes were washed three times in wash

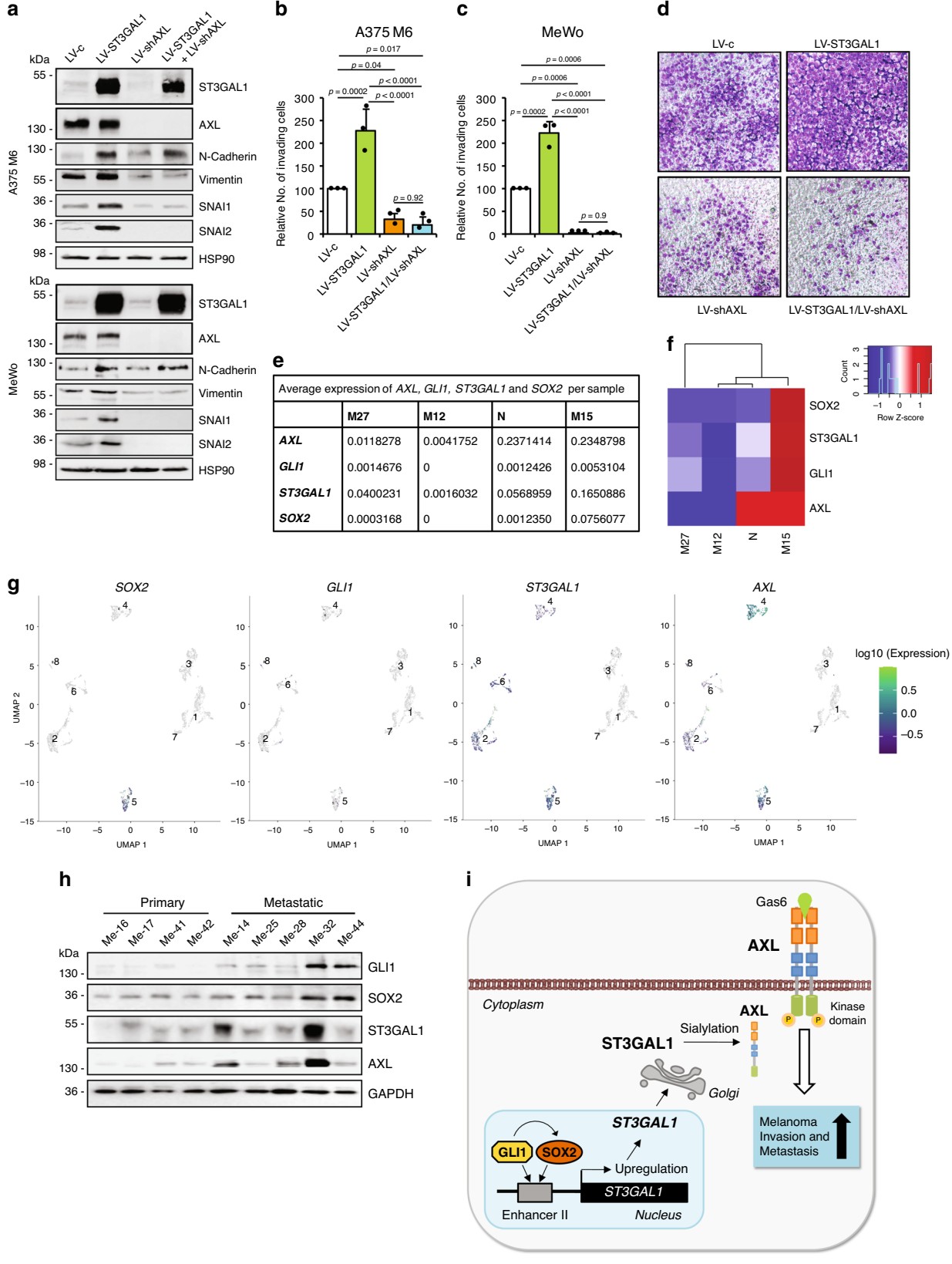

buffer (R&D System), incubated for 2 h with Anti-Phospho-Tyrosine-HRP Detection Antibody (R&D System) and imaged on the ChemiDoc MP Imaging System.

**Statistical analysis.** Data represent mean ± s.d or mean ± s.e.m. values calculated on at least three independent experiments. To assess normal distribution and homoscedasticity for each quantitative outcome in each group Kolmogorov–Smirnov's test and Bartlett's test was used respectively. Statistical significance was assessed by two-tailed unpaired Student's $t$ test, by ANOVA or Welch ANOVA and Tukey's, Dunnett's or Games-Howell's tests for multiple comparisons. For TCGA analysis statistical significance was determined using Welch's $t$-test. Details on the number of independent experiments or samples and statistical tests can be found in figure legends. $P$ value < 0.05 was considered statistically significant.

**Fig. 8 AXL is the main mediator of the pro-invasive effect of ST3GAL1. a** Western blot of ST3GAL1, AXL, and EMT markers in A375 M6 and MeWo cells transduced as indicated. HSP90 was used as loading control. **b, c** Matrigel invasion assays in A375 M6 and MeWo cells transduced as indicated. Data are represented as mean ± s.d. P value was calculated by ANOVA and Tukey's test ($n = 3$ biological independent experiments). **d** Representative images of A375 M6 as shown in **b**. **e** Average log-transformed normalized expression of *AXL, GLI1, SOX2,* and *ST3GAL1* in each sample. **f** Heat map of averaged expression of *SOX2, ST3GAL1, GLI1,* and *AXL* in normal human neonatal epidermal melanocytes (N) and in patient-derived xenografts (M27, M12, M15). **g** UMAP visualization of 3322 melanoma patient-derived xenograft cells and normal human neonatal epidermal melanocytes, colored by log-transformed normalized expression of SOX2, GLI1, ST3GAL1, and AXL respectively. Numbers in the plots indicate cell clusters. **h** Western blot of SOX2, GLI1, ST3GAL1, and AXL in primary and metastatic patient-derived melanoma cells. **i** Proposed mechanism of action of the SOX2/GLI1-ST3GAL1-AXL axis. ST3GAL1 is transcriptionally regulated by both GLI1 and SOX2 and drives melanoma cell invasion and metastasis. ST3GAL1 induces ligand-dependent AXL dimerization and consequent autophosphorylation at residue Tyr702. Blots in **a** and **h** are representative of $n = 3$ biological independent experiments. HSP90 and GAPDH were used as loading controls. Source data are provided as Source Data file.

**Reporting summary**. Further information on research design is available in the Nature Research Reporting Summary linked to this article.

## Data availability

RNAseq and scRNAseq primary data generated in this manuscript are available from GEO under accession numbers GSE159049 and GSE159597. The unique raw and normalized enriched MAL-bound proteins identified by MS are available as Supplementary Data 2. Public available microarray data used in this study were obtained from GSE46517, GSE7553, GDS1375, and GDS3966. All the data supporting this study are available within the article, the Supplementary file, the Source Data file, as indicated in the Reporting Summary for this article. A Reporting Summary for this article is available as a Supplementary Information file. Source data are provided with this paper.

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

## Acknowledgements

This work was supported by a grant from the Italian Association for Cancer Research (AIRC; IG-23091) and institutional funding from the Institute for Cancer Research, Prevention and Clinical Network (ISPRO) to B.S., by CA136526 to M.E.F.Z. and by K08CA215105 to A.M.; S.P. was supported by AIRC fellowship (project n. 21168). The authors wish to thank the Cassa di Risparmio di Padova e Rovigo (Cariparo) Holding for funding the acquisition of the LTQ-Orbitrap XL mass spectrometer.

## Author contributions

Experimental design: S.P. and B.S.; qPCR and western blots: S.P. and G.A.[1,2]; Generation of expression vectors: S.P.; In vitro assays: S.P., G.A.[1,2], and F.M.; Immunohistochemistry: S.P.[1,2] and G.A.[1,2]; ChIP and Luciferase reporter assays: S.P.; Clinical samples acquisition and pathological evaluation, histology, and lung metastasis evaluation: B.S.; Flow cytometry and CTC evaluation: C.S. and S.P.; Proteomic analysis: G.A.[8,9] and I.B.; PDX development and tissue processing: L.Y., B.D.P., J.N.S., and A.M.; RNA libraries, RNA-sequencing, scRNA-sequencing, and bioinformatic analysis: L.L.A., R.M.C., L.Y., C.Z., H.L., J.I.D., and M.E.F.Z.; Validation of MS and RNA-seq data: S.P.; Statistical analysis: L.T., B.D.P., and C.Z.; Manuscript writing: S.P., M.E.F.Z., and B.S. All authors provided intellectual input, reviewed, and edited the manuscript.

## Competing interests

The authors declare no competing interests.
