## [Peer Review File · Nature Communications]

Reviewers' comments:

Reviewer #1 (Expertise: Sialylation, cancer, Remarks to the Author):

This manuscript describes a novel Sox2/Gli1 axis that upregulates the ST3Gal1 sialyltransferase to promote melanoma invasiveness and metastasis. The study integrates a number of elements in a multi-step pathway linking Sox2/Gli to increased ST3Gal1 transcription and subsequently, sialylation of targets such as AXL and β 4 integrin. Furthermore the manuscript proposes that two drugs known to affect other targets can be repurposed to inhibit ST3Gal1 catalytic activity. The major strength of the manuscript is that the individual parts of the overall story (e.g., Sox2/Gli1 regulation of ST3Gal1; ST3Gal1 regulation of AXL, etc.) have novelty and potential significance. The major concern is that these individual elements in the multi-step pathway are not investigated or validated thoroughly enough. There are missing pieces of data that are crucial for supporting the overall pathway. Finally, the experiments regarding the activity of the inhibitors seem premature and should be deleted.

1) The quality of some of the blots needs improvement; in some instances the data are not highly convincing (Fig1a, Fig1g). Also it appears from Fig 1 that there may be reciprocal regulation of Sox2 and Gli (Sox2 knockdown downregulates Gli, and Gli knockdown suppresses Sox 2). This should be commented on in the manuscript.

2) While the manuscript shows that Sox2 and Gli1 knockdown suppresses ST3Gal1, there isn't much evidence that ectopic expression of Sox2 and Gli1 increases ST3Gal1 protein expression.

3) The ChIP experiment (Fig 2B) should be conducted with a second cell model (e.g., A375M6).

4) Fig 3: This figure should provide a greater comparison between normal and tumor tissues. For example, it is important to show ST3Gal1 staining in normal skin tissue and in the nevi (Fig 3d). Likewise, in Fig 3f, the melanoma cell lines should be evaluated with a normal melanocyte control line on the same blot. Additionally, it would be interesting to know whether the expression of Sox2, Gli1 and ST3Gal1 correspond in patient samples, or in melanoma cell lines (ie, are Sox2 and Gli high in the lines with high ST3Gal1 expression?).

5) The data in Supplementary Fig 2 are quite important, as they provide the functional link supporting a Sox2/Gli/ST3Gal1 signaling axis in tumor cell behavior. One wonders why these data were selected as supplementary rather than being included in the manuscript proper. Also, there are some questions about this Figure. In SFig2a, the strong overexpression of Sox2 did not appear to have much effect on ST3Gal1 expression. As an added note, it would be informative to examine EMT markers in these cell models.

6) Fig 6: In Fig6c, the blot for α 5 integrin shows a band at \sim 250 kD. This is not the correct size for this integrin. Fig 6d: the RTK assay should be verified by directly blotting for phospho-AXL in the cell lines with ectopic ST3Gal1 expression. These data could be included in Fig 6e. Also, EMT markers should be examined in the AXL KD cells.

7) Fig 7: There are a number of concerns regarding the inhibitor studies, and at present, this line of investigation seems premature. Specific suggestions:

(a) Flow cytometry should be performed with MAL-II to verify changes in surface α 2-3 sialylation (ie, the lectin precipitations for AXL and β 4 integrin are insufficient). Also, it is important to assess the effects of the drugs on both cell models (A375 and MeWo). As a side note, differences evoked by famotidine in AXL and β 4 integrin sialylation are minor.

(b) Flow cytometry should be conducted with SNA to assess whether α 2-6 sialylation is altered. One concern is that famotidine and fludarabine are acting broadly on multiple sialyltransferases.

(c) The most significant concern is that effects on migration/invasion are not related to ST3Gal1 at all, but are rather effects of the drugs on other known targets. Alternatively, there is a question about whether diminished invasion is secondary to toxicity of the drugs (of particular concern for fludarabine).

Reviewer #2 (Expertise: Sox2/Gli1, Remarks to the Author):

The manuscript by Pietrobono et al., describes their findings that a Sox2-Gli1 axis regulates the expression of the sialyltransferase ST3GAL1, which in turn sialylates the tyrosine kinase receptor Axl, promoting the invasion and metastasis of melanoma. They also demonstrate that two small molecule inhibitors of ST3GAL1 could abrogate the sialylation and arrest invasion and migration of melanoma cell lines. A series of biochemical and in vivo experiments have been conducted to support their hypothesis. The studies are overall interesting and well designed; at the same time, a certain number of additional experiments would be needed to make the story complete and fully convincing. The main comments are as follows:

1. The finding that Sox2 and Gli1 depletion both affects ST3GAL1 expression (Fig 1g) is interesting. While the ChIP assays show that both Gli1 and Sox2 binding sites contribute to the expression of ST3GAL1, it is not clear if there is a sequential regulation of expression, since it has been reported that Gli1 can regulate Sox2 expression. It would strengthen the data in Fig. 2 D-F, if Sox2 can be depleted and compensated with Gli1, and vice versa; similarly, it would provide more granularity if the transient transfection experiment in panel 2g is expanded to include depletion of Gli1 when Sox2 is overexpressed and vice versa.

2. Data in Figure 3 is interesting, especially 3d-f. Based on the data presented later in the manuscript, this data should ideally be supplemented with sequential IF and multi-spectral imaging or similar technique to see the levels of Sox2, Gli1, ST3GAL1 and pAxl in the same tumor sections. Is there a correlation between the levels of the antigens in the tumors? Data in Figure 3e does not reveal if

there is a higher proportion of metastatic cases with elevated ST3GAL1 compared to the primary tumors. If such information can be gleaned from the staining, it would provide a more compelling case for the invasion/migration experiments presented in Figure 4.

3. The biochemical data in Figure 4 is interesting. Given the observations with the ST3GAL1 inhibitors on cell viability, it would also be useful to assess if depletion or overexpression of ST3GAL1 affect cell viability or cell cycle progression. Does ST3GAL1 give a survival advantage to cells? This issue comes up later too.

4. The animal experiment is suggestive; it would have been more convincing if a related sialyltransferase which is not altered by Sox2/Gli1 is knocked out as well, as a control. Further, it would strengthen the data if ST3GAL1 is overexpressed in a Sox2 or Gli1 null cell line, to see if it could rescue tumor growth or metastasis. It is surprising that an actual tumor growth is not measured in a sub-q or orthotopic implantation experiment.

5. Identification of Axl as a target of ST3GAL1 is interesting and perhaps relevant. Given the hits presented in the Supplementary table 2, there are other tyrosine kinase receptors that appear to be sialylated, including EGFR and others. The lack of relevance of NGFR is shown; it would be good to examine EGFR as well, since there are earlier reports of its overexpression in melanoma. Further, the experiments where Axl is depleted, should also include NGFR and perhaps EGFR.

6. The IC50 data in the Supplementary Figure 5 should ideally be presented as a standard IC50 curve, with 10 data points.

7. Since the two agents appear to affect the invasive property of the cells, with some effect on viability, the translatability of these agents should be tested. Animal experiments would be the best option, but at a bare minimum, the utility of these agents to overcome BRAF inhibitor or Trametinib resistance should be examined in cell culture models. As it is presented now, the agents would be of limited translational utility.

8. The title of the study is not fully justified, unless additional proteins discovered in the MAL pull down experiment are ruled out experimentally.

9. It would be interesting and relevant to see if the overexpression of ST3GAL1 and activation of Axl is elevated in melanomas harboring different driver mutations; for example, what is the prevalence in B-Raf mutant melanoma versus the rest?

Overall, this study is interesting but can be improved significantly by a certain number of additional experiments.

Reviewer #3 (Expertise: Melanoma metastasis, AXL, Remarks to the Author):

Overall the manuscript is thorough, logical, well written and is of significance to the field. The authors have done a great job in investigating the mechanistic details underlying the role of ST3GAL1 in promoting melanoma progression. I commend the authors on their bioinformatic approaches, particularly with regard to their virtual drug screening. Overall, the in vitro data is very well done; however, there are some issues with their in vivo approach.

Major concerns:

Their main point of mechanism with regards to ST3GAL1 is via increasing melanoma cell invasion and migration. The in vivo approach of intracardiac injection is not a true indicator of metastatic progression, particularly from an invasion stand point. It is more indicative of how the melanoma cells are able to colonize (seed, survive and proliferate) in metastatic environments and gives no real indication of whether cells are more invasive within a primary tumor, and whether it increases the level of dissemination.

To truly test this, it would be better to form a primary tumor within a mouse and examine the levels of cells that are able to enter the blood stream (circulating tumor cells, CTCs) while also performing IHC to investigate micrometastasis in organs such as the lung. This can be accomplished via IHC analysis of SOX10 of parental melanoma cells, or they can create GFP/mCherry cells (or even use their luciferase cells) and perform IHC staining for these in the lung. Using fluorescent markers would also allow isolation of these cells from the blood to analyze CTCs. I feel this is an important experiment given that their main mechanism of action is invasion.

Furthermore, I feel that performing the alternate experiment where st3GAL1 is overexpressed in cells and metastasis is examined, should also be performed. If this does increase metastasis relative to a control, will treating the mice with the two drugs decrease progression?

Performing these experiments would greatly increase the impact of this research, particularly from a translational perspective, although I will understand if the editor believes that the drug treatment experiments are potentially not within the scope for this current study and more of a future proposal.

Minor issues:

In the results, it would be to describe in a sentence, the actual datasets you are referring to (eg TCGA) rather than just giving the geo code.

On page 5, your explanation of figure 3C needs grammatical correction.

Figure 3D needs to be labelled to identify which of these are nevi, primary or metastatic melanomas.

For figure 3E, briefly described how you quantified this data in the figure legend.

Figure 3F, which cell lines are metastatic and which are not? Label this on the figure.

Your assay in 6D needs to be explained more clearly in the results section. It is very hard to interpret with the way it has been written, particularly for those who have never heard of the assay. A brief description of what justifies activation or phosphorylation relative to control would help.

If you overexpress AXL in ST3GAL knockdown cells, do you rescue the invasive and migratory phenotype? This would suggest that sialylation is the key effector of this pathway if it didn't rescue invasion in these knockdown cells.

Are there any assays where you can study AXL dimerization or GAS6 binding in ST3GAL overexpressing cells? These assays would greatly enhance the mechanistic insight. Why not knockdown ST3GAL and treat with GAS6 and see if this decreases invasion relative to GAS6 treatment in control cells?

Some studies suggest AXL is not involved or doesn't correlate with invasion, particularly within melanoma. In the discussion, you should briefly mention that AXL and invasion is controversial and is seemingly context dependent in melanoma and give some examples.

POINT-BY-POINT RESPONSE

Reviewer 1:

1) The quality of some of the blots needs improvement; in some instances the data are not highly convincing (Fig1a, Fig1g).

Response: We have improved quality of Western blot in **Figure 1a** and **Figure 1g** to a level we hope the Reviewer finds acceptable.

2) Also it appears from Fig 1 that there may be reciprocal regulation of Sox2 and Gli (Sox2 knockdown downregulates Gli, and Gli knockdown suppresses Sox2). This should be commented on in the manuscript.

Response: We apologize for our omission. In the revised manuscript, as suggested, we have discussed the existence of a reciprocal interplay between SOX2 and GLI1 contributing to the transcriptional activation of ST3GAL1. Our data suggest that the regulation of ST3GAL1 by SOX2 is direct, whereas GLI1 acts both directly and indirectly via SOX2. This result is consistent with a previous report demonstrating that SOX2 is a target of GLI1 in melanoma (Santini *et al*, *Oncogene*, 2014 doi: 10.1038/onc.2014.71).

3) While the manuscript shows that Sox2 and Gli1 knockdown suppresses ST3Gal1, there isn't much evidence that ectopic expression of Sox2 and Gli1 increases ST3Gal1 protein expression.

Response: We performed new Western blot analyses in A375 M6 and SSM2c cells with ectopic expression of SOX2 or GLI1. Blots show that SOX2 or GLI1 (3-days post-selection with puromycin) consistently increases ST3GAL1 protein level (**New Figure 1h**). These new data further support our previous finding of a positive regulation of ST3GAL1 by SOX2 and GLI1.

4) The ChIP experiment (Fig 2B) should be conducted with a second cell model (e.g., A375M6).

Response: As requested, a ChIP experiment was conducted in a second cell model (A375 M6 cells) showing consistent results with those obtained in SSM2c cells as shown in **New Figure 6c** (previous Figure 2b). These data confirm that both SOX2 and GLI1 co-occupy ST3GAL1 distal enhancer element (enhancer II), with about 7-8 fold enrichment in ST3GAL1 signal by SOX2 and 3-4 fold enrichment by GLI1 (**Page 7**).

5) Fig 3: This figure should provide a greater comparison between normal and tumor tissues. For example, it is important to show ST3Gal1 staining in normal skin tissue and in the nevi (Fig 3d). Likewise, in Fig 3f, the melanoma cell lines should be evaluated with a normal melanocyte control line on the same blot.

Response: We thank the Reviewer for the suggestion. In the revised manuscript we now show that the expression of ST3GAL1 in human normal skin and nevi is very low compared to the tumor samples (**New Figure 2d and e**). Consistently, Western blot shows that normal human epidermal melanocytes (NHEM) display very low ST3GAL1 protein expression, whereas melanoma cells display variable levels of ST3GAL1 (**New Figure 2f**).

6) Additionally, it would be interesting to know whether the expression of Sox2, Gli1 and ST3Gal1 correspond in patient samples, or in melanoma cell lines (ie, are Sox2 and Gli high in the lines with high ST3Gal1 expression?).

Response: To address the question about the correlation between SOX2, GLI1 and ST3GAL1 in melanoma, we first tested the expression of SOX2, GLI1 and ST3GAL1 mRNA in melanoma cell lines. Quantitative PCR showed a positive correlation only between SOX2 and ST3GAL1 transcripts (**New Supplementary Figure 6**). Furthermore, to investigate if SOX2, GLI1, ST3GAL1 and AXL are co-expressed in melanoma, we performed single cell RNA-sequencing analysis in cells derived from melanoma brain metastasis PDX models (M12, M15, M27) as well as normal melanocyte cultures (**New Supplementary Figure 10 and 11**). These data show that ST3GAL1, GLI1, SOX2 and AXL are co-expressed in melanoma cells with similar or higher levels in tumor

samples (M) vs. normal (N), except M12 (**New Figure 8e-g**). Consistently, Western blot analysis of patient-derived short-term melanoma cultures shows that primary melanoma cells express lower levels of SOX2, GLI1, ST3GAL1 and AXL compared to metastatic melanoma cells and that in the latter the components of the SOX2/GLI1-ST3GAL1-AXL axis are co-expressed (**New Figure 8h**). These data are discussed in the manuscript (**Page 10**).

7) The data in Supplementary Fig 2 are quite important, as they provide the functional link supporting a Sox2/Gli/ST3Gal1 signaling axis in tumor cell behavior. One wonders why these data were selected as supplementary rather than being included in the manuscript proper. Also, there are some questions about this Figure. In SFig2a, the strong overexpression of Sox2 did not appear to have much effect on ST3Gal1 expression. As an added note, it would be informative to examine EMT markers in these cell models.

Response: As suggested by the Reviewer, we moved data regarding the functional link supporting SOX2/GLI1-ST3GAL1 axis, namely the requirement of ST3GAL1 for SOX2/GLI1-mediated invasion, in **New Figure 6i-k**, just below the molecular mechanism (previous Figure 2). The experiment showing that ST3GAL1 ectopic expression rescues the reduced invasion of SOX2- or GLI1-depleted melanoma cells is reported in **New Supplementary Figure 5 i-k**. In addition, we re-made Western blots and performed new qPCR. These new data show that SOX2 overexpression clearly increases ST3GAL1 expression, both at protein (**New Figure 6j**) and mRNA levels (**New Supplementary Figure 5c and d**). As suggested by the Reviewer, we examined EMT markers, including N-cadherin, SNAI1 and SNAI2, in these cell models. The results are part of **New Figure 6j** and **New Supplementary Figure 5j** and show that ST3GAL1 acts as a mediator of SOX2 and GLI1 in melanoma cell invasion by positively modulating the transcription factors SNAI1 and SNAI2.

8) Fig 6: In Fig6c, the blot for $\alpha 5$ integrin shows a band at ~250 kD. This is not the correct size for this integrin.

Response: We apologize for mistake, we now corrected it (**New Figure 7c**).

9) Fig 6d: the RTK assay should be verified by directly blotting for phospho-AXL in the cell lines with ectopic ST3Gal1 expression. These data could be included in Fig 6e. Also, EMT markers should be examined in the AXL KD cells.

Response: As suggested by the Reviewer, we verified the RTK assay by blotting three different melanoma cell lines (A375 M6, MeWo and M51) overexpressing ST3GAL1 with an anti-PAXL antibody. These data show that ectopic expression of ST3GAL1 in melanoma cells increases phosphorylation of AXL at residue Tyr702 in the catalytic kinase domain (**New Figure 7f**), which becomes autophosphorylated in response to AXL activation, confirming that sialylation of AXL by ST3GAL1 promotes its phosphorylation and activation. The involvement of AXL was also confirmed by immunoprecipitation followed by immunoblotting with anti-phospho-Tyrosine antibody (**New Figure 7e**). Finally, we examined the expression of EMT markers in A375M6 and MeWo AXL-silenced cells (**New Figure 8a**). Upregulation of SNAI1 and SNAI2 by ST3GAL1 is abrogated upon AXL silencing, confirming the role of AXL as the main mediator of ST3GAL1-induced invasion.

10) Fig 7: There are a number of concerns regarding the inhibitor studies, and at present, this line of investigation seems premature. Specific suggestions:

(a) Flow cytometry should be performed with MAL-II to verify changes in surface $\alpha 2-3$ sialylation (ie, the lectin precipitations for AXL and $\beta 4$ integrin are insufficient). Also, it is important to assess the effects of the drugs on both cell models (A375 and MeWo). As a side note, differences evoked by famotidine in AXL and $\alpha 4$ integrin sialylation are minor.

(b) Flow cytometry should be conducted with SNA to assess whether $\alpha 2-6$ sialylation is altered. One concern is that famotidine and fludarabine are acting broadly on multiple sialyltransferases.

(c) The most significant concern is that effects on migration/invasion are not related to ST3Gal1 at all, but are rather effects of the drugs on other known targets. Alternatively, there is a question

about whether diminished invasion is secondary to toxicity of the drugs (of particular concern for fludarabine).

Response: We thank the Reviewer for the constructive comments, following her/his suggestion shared also by the Editorial team, we have removed the ST3GAL1 inhibitor study.

Reviewer 2:

1. The finding that Sox2 and Gli1 depletion both affects ST3GAL1 expression (Fig 1g) is interesting. While the ChIP assays show that both Gli1 and Sox2 binding sites contribute to the expression of ST3GAL1, it is not clear if there is a sequential regulation of expression, since it has been reported that Gli1 can regulate Sox2 expression. It would strengthen the data in Fig. 2 D-F, if Sox2 can be depleted and compensated with Gli1, and vice versa; similarly, it would provide more granularity if the transient transfection experiment in panel 2g is expanded to include depletion of Gli1 when Sox2 is overexpressed and vice versa.

Response: We thank the Reviewer for these comments. A previous study from our group demonstrated that GLI1 directly regulates the expression of SOX2 (Santini *et al*, *Oncogene* 2014 doi: 10.1038/onc.2014.71). Similarly, in this study we show that SOX2 modulates GLI1 expression (**New Figure 1g and h**), suggesting a reciprocal positive cross-regulation. To assess whether the observed induction of ST3GAL1 transcription could depend on a sequential regulation of the two transcription factors, we overexpressed SOX2 in cells depleted for GLI1 and vice versa. Luciferase assays showed that SOX2 overexpression partially compensates for the absence of GLI1 (**New Supplementary Figure 5a**), similarly to what happens after ectopic expression of GLI1 in SOX2-silenced cells (**New Supplementary Figure 5b**), suggesting that both GLI1 and SOX2 are required for the induction of ST3GAL1 transcription. This is also confirmed at both mRNA and protein levels (**New Fig. 6g; New Supplementary Figure 5c-f**).

To assess the contribution of SOX2 or GLI1 on ST3GAL1 transactivation, we transiently transfected GLI1 in SOX2-silenced melanoma cells and SOX2 in GLI1-silenced melanoma cells. The transactivation of ST3GAL1 GBSmut enhancer II by GLI1 (**New Figure 6h**) is completely abrogated when GLI1 is transiently overexpressed in cells silenced for SOX2 (**New Supplementary Figure 5g**), similarly to that of SBS2mut enhancer II by SOX2 (**New Supplementary Figure 5h**). However, while SOX2-induced transcriptional activation of ST3GAL1 occurs even in absence of a functional GBS, disruption of SBS2 appears to reduce transactivation by GLI1 (**New Figure 6h**). This effect is still abrogated when GLI1 is transiently overexpressed in SOX2-silenced cells (**New Supplementary Figure 5g**). These new data, discussed in the manuscript (**Pages 7-8 and 12**), suggest that the transcriptional regulation of ST3GAL1 by SOX2 is direct, whereas GLI1 acts both directly and indirectly via SOX2.

2. Data in Figure 3 is interesting, especially 3d-f. Based on the data presented later in the manuscript, this data should ideally be supplemented with sequential IF and multi-spectral imaging or similar technique to see the levels of Sox2, Gli1, ST3GAL1 and pAxl in the same tumor sections. Is there a correlation between the levels of the antigens in the tumors? Data in Figure 3e does not reveal if there is a higher proportion of metastatic cases with elevated ST3GAL1 compared to the primary tumors. If such information can be gleaned from the staining, it would provide a more compelling case for the invasion/migration experiments presented in Figure 4.

Response: To investigate if SOX2, GLI1, ST3GAL1 and AXL are co-expressed in melanoma, we performed single cell RNA-sequencing analysis in cells derived from melanoma brain metastasis PDX models (M12, M15, M27) as well as normal melanocyte cultures (**New Supplementary Figure 10 and 11**). These data show that *ST3GAL1*, *GLI1*, *SOX2* and *AXL* are co-expressed in melanoma cells with similar or higher levels in tumor samples (M) vs. normal (N), except M12 (**New Figures 8e-g**). Consistently, Western blot analysis of patient-derived short-term melanoma cultures shows that primary melanoma cells express lower levels of SOX2, GLI1, ST3GAL1 and AXL compared to metastatic melanoma cells and that in the latter the components of the SOX2/GLI1-ST3GAL1-AXL

axis are co-expressed (**New Figure 8h**). These data are discussed in the manuscript (**Page 10**).

Furthermore, the expression of ST3GAL1 was analyzed in additional melanoma metastatic samples by IHC and data are now expressed as percentage of cases. As shown in the **New Figure 2e** (previous Figure 3e), there is higher proportion of metastatic cases with medium/high ST3GAL1 staining compared to primary melanomas and nevi, supporting the association between increased ST3GAL1 expression and melanoma progression (**Pages 5 and 11**).

3. The biochemical data in Figure 4 is interesting. Given the observations with the ST3GAL1 inhibitors on cell viability, it would also be useful to assess if depletion or overexpression of ST3GAL1 affect cell viability or cell cycle progression. Does ST3GAL1 give a survival advantage to cells? This issue comes up later too.

Response: Modulation of ST3GAL1 has minor effects on melanoma cell growth *in vitro*. Both ST3GAL1 shRNAs slightly decreases the number of viable cells and increased the early apoptotic fraction of cells compared to non-targeting control (LV-c) by affecting BAX/BCL2 ratio, whereas ST3GAL1 overexpression leads to a modest increase in cell growth and survival (**New Supplementary Fig. 2**). In addition, ST3GAL1 depletion did not affect melanoma cell cycle progression.

Cell cycle analysis of A375 M6 and MeWo cells transduced as indicated. Data are shown as mean \pm s.e.m.

4. The animal experiment is suggestive; it would have been more convincing if a related sialyltransferase which is not altered by Sox2/Gli1 is knocked out as well, as a control. Further, it would strengthen the data if ST3GAL1 is overexpressed in a Sox2 or Gli1 null cell line, to see if it could rescue tumor growth or metastasis. It is surprising that an actual tumor growth is not measured in a sub-q or orthotopic implantation experiment.

Response: To address the question of the involvement of a related sialyltransferase which is not altered by SOX2/GLI1, we analyzed the role of ST3GAL4, an α -2,3-sialyltransferase which is not reported in the list of SOX2/GLI1-regulated genes (**Supplementary Table 1**) and is among the proteins sialylated by ST3GAL1 (**Supplementary Table 3**). Functional data indicate that silencing of ST3GAL4 has a minor effect on reducing melanoma cell invasion compared to ST3GAL1. In addition, depletion of ST3GAL4 does not affect the increase in melanoma cell invasion induced by ST3GAL1, suggesting a minor role for this sialyltransferase. As for animal experiments, the national regulations of Italian Ministry of Health require a very strict 3R (replacement, reduction, refinement) and do not approve *in vivo* experiments in absence of strong and clear effects *in vitro*.

a) Quantitative real-time PCR (qPCR) analysis of ST3GAL1 and ST3GAL4 in A375 M6 and MeWo cells transduced as indicated. b,c) Relative number of invading cells in A375 M6 (b) and MeWo (c) cells transduced as indicated. d) Relative number of invading cells in A375 M6 cells transduced with LV-c, LV-shST3GAL4 or LV-shST3GAL1. Data are shown as mean \pm s.e.m.

Regarding the overexpression of ST3GAL1 in a SOX2 or GLI1 null cell line, we made several attempts to knock-out GLI1 or SOX2 in cancer cell lines using the CRISPR/Cas9 technology. Apparently, most cancer cells do not tolerate complete ablation of GLI1. We obtained U87 (glioblastoma) and M51 (melanoma) GLI1-null cell lines and SK-Mel-28 SOX2-null cell lines, but unfortunately these cell lines were not able to efficiently give rise to metastases *in vivo*. More importantly, ST3GAL1 does not rescue the reduced melanoma cell growth induced by depletion of SOX2 and GLI1 *in vitro*.

Growth curve in SSM2c cells transduced as indicated. Data are shown as mean \pm s.e.m.

Furthermore, ST3GAL1 modulation does not significantly affect orthotopic tumor growth *in vivo* (New Figure 5c). Altogether, these data do not support the premises to conduct an *in vivo* experiment where ST3GAL1 is overexpressed in absence of SOX2 or GLI1 (based on the 3R of Ministry of Health).

5. Identification of Axl as a target of ST3GAL1 is interesting and perhaps relevant. Given the hits presented in the Supplementary table 2, there are other tyrosine kinase receptors that appear to be sialylated, including EGFR and others. The lack of relevance of NGFR is shown; it would be good to examine EGFR as well, since there are earlier reports of its overexpression in melanoma. Further, the experiments where Axl is depleted, should also include NGFR and perhaps EGFR.

Response: The Reviewer is correct in saying that other tyrosine kinase receptors appear to be sialylated by ST3GAL1, including EGFR and NGFR. We present several lines of evidence that rule out the involvement of NGFR and EGFR downstream of ST3GAL1. First, ectopic expression of ST3GAL1 increases phosphorylation status of AXL compared to scrambled LV-c cells, without altering phosphorylation of EGFR or NGFR (**New Figure 7d**). Second, this was confirmed by immunoprecipitation of AXL, EGFR or NGFR followed by immunoblotting with anti-phospho-Tyrosine (**New Figure 7e**). Third, silencing of AXL in melanoma cells is able to fully revert the pro-invasive effects mediated by ST3GAL1 (**New Figure 8a-d**). Conversely, depletion of NGFR or EGFR in cells overexpressing ST3GAL1 did not affect ST3GAL1-dependent cell invasiveness (**new Supplementary Figure 7 b-g**). AXL impacts on the migratory ability of A375 M6 ST3GAL1-overexpressing cells but not that of MeWo cells, suggesting that additional mediators could be involved in this process, possibly NGFR (**New Supplementary Figure 9; Supplementary Figure 7h, i**). Overall, our data suggest that AXL is the major mediator of the pro-invasive role played by ST3GAL1 in melanoma and are discussed in the manuscript (**Pages 9 and 13**).

6. The IC50 data in the Supplementary Figure 5 should ideally presented as a standard IC50 curve, with 10 data points.

Response: Following the Editorial team's suggestion, we removed all the data related to the ST3GAL1 inhibitor studies from the current manuscript.

7. Since the two agents appear to affect the invasive property of the cells, with some effect on viability, the translatability of these agents should be tested. Animal experiments would be the best option, but at a bare minimum, the utility of these agents to overcome BRAF inhibitor or Trametinib resistance should be examined in cell culture models. As it is presented now, the agents would be of limited translational utility.

Response: See response to point 6.

8. The title of the study is not fully justified, unless additional proteins discovered in the MAL pull down experiment are ruled out experimentally.

Response: Our experimental data appear to rule out the involvement of NGFR and EGFR in the ST3GAL1-mediated melanoma cell invasiveness, but we cannot exclude the possibility that other RTK or proteins can mediate ST3GAL1 functions. Therefore, the title was slightly modified to "*ST3GAL1, a novel target of the SOX2-GLI1 transcriptional complex, promotes melanoma metastasis in part through the activation of AXL*".

9. It would be interesting and relevant to see if the overexpression of ST3GAL1 and activation of Axl is elevated in melanomas harboring different driver mutations; for example, what is the prevalence in B-Raf mutant melanoma versus the rest?

Response: To address this question we correlated the expression of *ST3GAL1* and *AXL* to the mutational status of BRAF, PTEN, p53 and NRAS, KRAS and HRAS. Analysis of a cohort of 481 human melanoma samples from The Cancer Genome Atlas (TCGA) shows that expression of *ST3GAL1* was significantly increased in *BRAF* mutant melanomas, whereas that of *AXL* in TP53 mutant melanomas compared to wild type cases (**New Supplementary Figure 12**), suggesting the existence of both non-genetic (the GLI1-SOX2 transcriptional mechanism) and genetic mechanisms controlling the ST3GAL1-AXL axis (**Page 10**).

Reviewer 3:

1) Their main point of mechanism with regards to ST3GAL1 is via increasing melanoma cell invasion and migration. The in vivo approach of intracardiac injection is not a true indicator of metastatic progression, particularly from an invasion stand point. It is more indicative of how the melanoma cells are able to colonize (seed, survive and proliferate) in metastatic environments and

gives no real indication of whether cells are more invasive within a primary tumor, and whether it increases the level of dissemination.

To truly test this, it would be better to form a primary tumor within a mouse and examine the levels of cells that are able to enter the blood stream (circulating tumor cells, CTCs) while also performing IHC to investigate micrometastasis in organs such as the lung. This can be accomplished via IHC analysis of SOX10 of parental melanoma cells, or they can create GFP/mCherry cells (or even use their luciferase cells) and perform IHC staining for these in the lung. Using fluorescent markers would also allow isolation of these cells from the blood to analyze CTCs. I feel this is an important experiment given that their main mechanism of action is invasion. Furthermore, I feel that performing the alternate experiment where ST3GAL1 is overexpressed in cells and metastasis is examined, should also be performed. If this does increase metastasis relative to a control, will treating the mice with the two drugs decrease progression? Performing these experiments would greatly increase the impact of this research, particularly from a translational perspective.

Response: To address whether modulation of ST3GAL1 is involved in metastasis formation and progression *in vivo*, we used a xenograft model of metastasis. We performed orthotopic injection of highly metastatic melanoma cells (A375 M6) stably transduced with GFP/luciferase reporter and LV-shST3GAL1.1, LV-ST3GAL1 (LV-ST3GAL1) or scramble control (LV-c), and monitored mice for:

- local/primary tumor growth;
- presence of circulating melanoma cells in the blood stream (CTCs);
- metastasis formation in the lungs.

We observed no significant difference in tumor growth between LV-c, LV-shST3GAL1.1 and LV-ST3GAL1, except a slight tumor growth delay in LV-shST3GAL1 mice (**New Figure 5c**). When local tumors reached the same tumor volume (**New Figure 5c and d**), primary tumors were surgically resected, and the ability of cells to enter blood stream (CTCs) and give rise to distant metastases was examined. Silencing of ST3GAL1 substantially reduced the frequency of circulating melanoma cells in the blood (**New Figure 5e**) and strongly inhibited the metastatic potential of A375 M6 cells (**New Figures 5f and g**). Mice injected with LV-shST3GAL1 exhibited significantly reduced number of lung micrometastases compared to LV-c (**New Figure 5h and i**). Ectopic expression of ST3GAL1 did not affect the number of CTCs nor number of micrometastases per lung compared to control (**New Figures 5e-i**), although mice injected with LV-ST3GAL1 exhibited higher number of macrometastases (**New Supplementary Figure 3**). These data provide evidence that ST3GAL1 plays a major role in the ability of aggressive melanoma cells to enter the blood stream, colonize distal organs and seed, survive and proliferate in the metastatic environment. These data reinforce our previous findings and are discussed in the manuscript (**Pages 6-7 and 11-12**).

2) Although I will understand if the editor believes that the drug treatment experiments are potentially not within the scope for this current study and more of a future proposal.

Response: Following the Editorial team's suggestion, we removed the ST3GAL1 inhibitor study from the present manuscript, because not within the scope for this current study but more for a future proposal.

3) In the results, it would be to describe in a sentence, the actual datasets you are referring to (eg TCGA) rather than just giving the geo code.

Response: As requested by the Reviewer, we describe the transcriptomic datasets used for the analysis in the Results section (**Pages 4-5**).

4) On page 5, your explanation of figure 3C needs grammatical correction.

Response: We made grammatical correction as suggested (**New Figure 2**).

5) Figure 3D needs to be labelled to identify which of these are nevi, primary or metastatic melanomas.

Response: We labelled **New Figure 2d** (previous Figure 3d) to identify normal skin, nevi, primary and metastatic melanomas.

6) For figure 3E, briefly described how you quantified this data in the figure legend.

Response: ST3GAL1 IHC staining was evaluated blindly and scored as negative (no signal), low, medium or high. This is now reported in New Figure 2 legend. Examples of negative, low, medium and high ST3GAL1 expression are included in **New Figure 2d** (previous Figure 3d), and presented as percentage of total cases.

7) Figure 3F, which cell lines are metastatic and which are not? Label this on the figure.

Response: We apologize for this omission. All cell lines reported in **New Figure 2f** (previous Figure 3f) are metastatic. In **New Figure 8h** we reported expression of SOX2, GLI1, ST3GAL1 and AXL in patient-derived short-term cultures from 4 primary and 5 metastatic melanomas.

8) Your assay in 6D needs to be explained more clearly in the results section. It is very hard to interpret with the way it has been written, particularly for those who have never heard of the assay. A brief description of what justifies activation or phosphorylation relative to control would help.

Response: As requested by the Reviewer, we have expanded the write-up of the results of the phospho RTK assay to make clearly the description of these findings (**Page 9**).

9) If you overexpress AXL in ST3GAL knockdown cells, do you rescue the invasive and migratory phenotype? This would suggest that sialylation is the key effector of this pathway if it didn't rescue invasion in these knockdown cells.

Response: To address this comment, we overexpressed AXL in ST3GAL1-silenced A375 M6 and MeWo cells. Our data show that overexpression of AXL does not rescue the invasive phenotype in absence of endogenous ST3GAL1 (**New Supplementary Figures 8a-c**). Similar results were obtained when AXL was activated by stimulation with GAS6 ligand (**New Supplementary Figure 8d and e**). These data reinforce the evidence that sialylation represents the key effector of this pathway.

10) Are there any assays where you can study AXL dimerization or GAS6 binding in ST3GAL overexpressing cells? These assays would greatly enhance the mechanistic insight. Why not knockdown ST3GAL and treat with GAS6 and see if this decreases invasion relative to GAS6 treatment in control cells?

Response: As suggested, we stimulated serum-starved cells with the Gas6 ligand and analyzed AXL dimerization in melanoma cells. While AXL appears to exist mainly as monomer, stimulation with Gas6 induced its partial dimerization only in presence of endogenous ST3GAL1 (**New Figure 7g**). Consistent with the dimerization result, A375 M6 cells depleted for ST3GAL1 showed no AXL phosphorylation at Tyr702 in response to ligand stimulation (**New Figure 7h**). These data support the requirement of ST3GAL1 for ligand-dependent AXL dimerization and consequent autophosphorylation at residue Tyr702, suggesting that sialylation might be responsible for increasing the affinity of AXL for its ligand Gas6. These data are discussed in the manuscript (**Pages 9 and 13**).

11) Some studies suggest AXL is not involved or doesn't correlate with invasion, particularly within melanoma. In the discussion, you should briefly mention that AXL and invasion is controversial and is seemingly context dependent in melanoma and give some examples.

Response: As requested by the Reviewer, in the Discussion we have included comments about the potential dual regulatory function of AXL on melanoma invasion (**Page 13**).

REVIEWERS' COMMENTS:

Reviewer#1:

The revised manuscript addresses all of my concerns

Reviewer#2:

The authors have added a significant amount of additional information and high quality data to support their original claims. Though certain experiments they have done to address certain issues are different from the ones suggested (for example, single-cell RNA seq in lieu of multi-spectral imaging to show co-expression), they have alleviated all the concerns I had raised in the prior version. The results do support their original hypothesis and I have no additional concerns.

Reviewer#3:

The authors have done an impressive job responding to the concerns addressed. The paper is acceptable for publication.